# MECHANISTIC INTERPRETATIONS AT MULTIPLE SCALES OF ABSTRACTION: REVISITING MODULAR ADDITION

## ABSTRACT

Prior work in mechanistic interpretability has analyzed how neural networks solve modular arithmetic tasks, but conflicting interpretations have emerged, questioning the universality hypothesis—that similar tasks lead to similar learned circuits. Revisiting modular addition, we identify that these discrepancies stem from overly granular analyses, which obscure the higher-level patterns that unify seemingly disparate solutions. Using a multi-scale approach—microscopic (neurons), mesoscopic (clusters of neurons), and macroscopic (entire network)—we show that all scales align on (approximate) cosets and implement an abstract algorithm resembling the approximate Chinese Remainder Theorem. Additionally, we propose a model where networks aims for a constant logit margin, predicting $\mathcal{O}(\log(n))$ frequencies—more consistent with empirical results in networks with biases, which are more expressive and commonly used in practice, than the $\frac{n-1}{2}$ frequencies derived from bias-free networks. By uncovering shared structures across setups, our work provides a unified framework for understanding modular arithmetic in neural networks and generalizes existing insights to broader, more realistic scenarios.

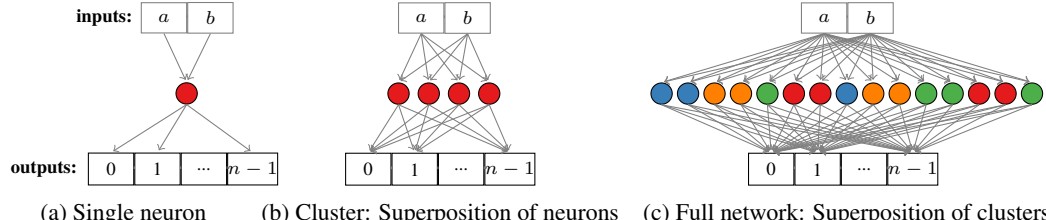

(a) Single neuron      (b) Cluster: Superposition of neurons      (c) Full network: Superposition of clusters

Figure 1: **Analysis at three levels of abstraction:** (a) per-neuron analysis reveals known algorithms in the literature (b) inter-neuron interactions within a cluster reveal cosets, and (c) full network with all clusters and inter-cluster interactions reveals an algorithm operating on cosets reminiscent of the Chinese Remainder Theorem. (Neuron color indicates cluster membership.)

## 1 INTRODUCTION

The *universality hypothesis* (Olah et al., 2020) is the claim that 'Analogous features and circuits form across models and tasks.' This hypothesis lays at the foundation of current efforts in mechanistic interpretability because it gives hope that reverse engineering neural networks will reveal interpretations that can be applied to a large class of networks. Mechanistic interpretability aims to uncover the precise mechanisms by which models transform inputs into outputs, offering insights into how specific neurons, layers, or circuits contribute to a network's overall function. Understanding how these networks organize information, form abstractions, perform computations, and make predictions is essential to building more trustworthy and reliable systems; moreover, such findings can guide the development of theory.

We build on the work of Nanda et al. (2023a), Chughtai et al. (2023b), Stander et al. (2023), Zhong et al. (2024), which mechanistically studied and reverse-engineered neural networks trained on modular addition and other group operations. These works found conflicting algorithms–either due to

random variation in the training process or human interpretation itself–eventually raising doubts about the universality of neural network representations when trained on group-theoretic tasks. It is necessary to walk through the history of this problem in order to situate our work. The study of modular addition by Nanda et al. (2023a) was an attempt to mechanistically explain the surprising "grokking" phenomenon observed in Power et al. (2022), where algorithmic tasks were found to suddenly generalize long after memorizing the training set. By reverse engineering a model trained to grok modular addition, Nanda et al. (2023a) discovered that it was possible to specify the computations performed by a specific network in a human-understandable way. This key result opened the door to a flood of research on the topic. Initially, Nanda et al. (2023a) identified a 'Fourier multiplication' algorithm, which later was generalized to the more general 'Group Composition via Representations' (GCR) framework (Chughtai et al., 2023b), *claimed to be universal for all groups*. However, subsequent studies have challenged this universality. Networks trained with different hyperparameters sometimes learned entirely different algorithms (e.g. The Clock, The Pizza, or made use of Lissajous-like curves) (Zhong et al., 2024). More definitively, Stander et al. (2023)–using the exact same experimental setup as Chughtai et al. (2023b)–showed that GCR fails to describe networks trained on the symmetric group, where coset-based structures (circuits) emerged instead. Consequently, Stander et al. (2023) and Zhong et al. (2024) provide evidence that even on something as simple as finite groups, there may be no universality in the algorithms neural networks learn.

This paper revisits the following fundamental question in mechanistic interpretability: do neural networks trained on group-theoretic tasks, such as modular addition, converge to universal structures, and if so, what form do these structures take? Focusing on modular addition, we embark on a classification of network structures, with the aim to find a scale at which studying the network makes our interpretation robust to changes in hyperparameters and random initializations. Our analysis reveals that regardless of whether the modulus is prime or composite, neural networks may be implementing a "macroscopic" algorithm for modular addition through a process resembling the Chinese Remainder Theorem, where neurons are organized into superimposed clusters representing (approximate) cosets of (approximate) subgroups. Within the context of modular addition, this macroscopic view both accounts for the variation seen in Nanda et al. (2023a) and Zhong et al. (2024) and also naturally draws a connection with the coset structures identified in the symmetric group (Stander et al., 2023), which previously disproved GCR's (Chughtai et al., 2023b) claims of universality. While we do not claim to have established universality across all groups, our findings restore hope that a coherent, abstract mechanism may exist. By identifying coset structures in modular addition, we link and reconcile previously conflicting findings, suggesting that networks may indeed learn universal structures at the appropriate level of abstraction.

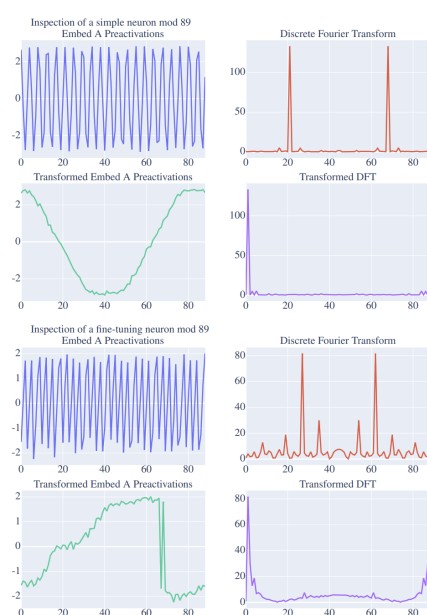

Figure 2: Comparing a simple neuron and fine-tuning neuron before and after transformation by a group isomorphism. The fine-tuning neuron has its DFT concentrating strongest on (27, 35, 19).

**Summary of contributions. 1)** We reverse engineer networks trained on $a + b \mod n$ for $n$ prime and composite, deriving an explanation robust to observations at three levels of abstraction: *neuron by neuron, clusters of neurons, and the entire network at once* (Fig. 1). As far as we are aware, composites have not been previously studied in detail. We conduct extensive experiments on the embeddings, preactivations, postactivations and contributions of neurons to the logits. Additionally, we study the distributions of what's learned (frequencies, phases, etc.) in the network over 100k seeds in order to identify general structures used by networks learning modular addition. These identified structures correspond to two classes of neurons we call "simple" and "fine-tuning" neurons. Simple neurons, in particular, play a key role in accounting for the variation in algorithms seen across the literature. See Fig. 2 for the difference between the two.

**2)** We construct a mathematical model for simple neurons allowing us to develop a more general framework than previous theoretical models that assume the network has no biases. This mathematical model breaks down the network from embeddings, to neurons, to logits in such a way that it accounts for and generalizes previous work. The values we observe in the embeddings and pre-activations come from projections of representations. When a single neuron fires strongly on two different inputs, we can interpret these inputs as being in a common "approximate coset". In clustering neurons and studying their contributions to logits, we find values consistent with an algorithm that uses approximate cosets, and is able to reproduce features of both clock and pizza algorithms. This is the first time the role of cosets has been identified in modular addition, which was, until now, thought to make use of seemingly unrelated algorithms–*i.e.* GCR (the clock) and the Pizza algorithms, thus connecting the literature under the umbrella of cosets.

**3)** We study the aforementioned clusters of neurons and give a toy model for how with superposition, *i.e.* the network using linear combinations of neurons, the neural network efficiently expresses the function. This model predicts $\mathcal{O}(\log n)$ frequencies (clusters), which matches experimental results in the literature, see Fig 5b, and improves on the previous model which uses $\mathcal{O}(n)$ (Gromov, 2023).

## 2 RELATED WORK

**Mechanistic interpretations of group-theoretic tasks, grokking and universality.** Popularized by works such as Cammarata et al. (2020), mechanistic interpretability has not only studied toy settings like Nanda et al. (2023b); Quirke et al. (2023); Zhang et al. (2023), but even tackled frontier models and shed light into the origins of in-context learning (Olsson et al., 2022). The universality hypothesis (Olah et al., 2020), (Li et al., 2015) asserts that different neural networks trained on similar tasks will converge to similar features and circuits; it is a key open question in the field of mechanistic interpretability. A common thread in attempts to interpret neural networks is the crucial need for caution and precision in the interpretation process (Adebayo et al., 2018; Bolukbasi et al., 2021; Sun et al., 2023; Poursabzi-Sangdeh et al., 2021; Jain & Wallace, 2019; Doshi-Velez & Kim, 2017). This has led to attempts to automate circuit discovery (Conmy et al., 2023) and understand the dynamics of circuit formation during training (Hoogland et al., 2024)

**Learning mathematical tasks.** Mathematical tasks are valuable testbeds for studying neural networks due to their structural sparsity, which isolates key variables and simplifies analysis, and the rich theoretical frameworks available to align human intuition with model representations. These tasks have driven progress in understanding network properties, representations, and dynamics (Elhage et al., 2022). Modular addition, the focus of this paper, has been extensively studied: Liu et al. (2022) link grokking to structured representations; Doshi et al. (2023) analyze generalizable representations under corrupted labels; Ding et al. (2024) model interactions of competitive and cooperative representations as linear differential equations; and He et al. (2024) explore the emergence of in-context learning and skill composition in modular arithmetic.

**Theoretical models of grokking modular addition.** Previous work, such as Gromov (2023), analyzed two-layer quadratic networks without biases for modular arithmetic, showing that solutions rely on *Fourier features* and deterministically use all $(n-1)/2$ frequencies. Morwani et al. (2024) extended this by proving such solutions maximize the margin, while Mohamadi et al. (2023) linked margin maximization to the grokking phenomenon in similar bias-free networks. In contrast, our primarily empirical work, supported by a heuristic theoretical model, incorporates biases and matches experimental observations, showing that biased networks use only $\mathcal{O}(\log(n))$ frequencies to solve modular addition, highlighting the subtleties of biases both in theory and in training dynamics for this problem.

## 3 PRELIMINARIES

### 3.1 GROUP (REPRESENTATION) THEORY AND THE CHINESE REMAINDER THEOREM

A **group** $(G, \circ)$ is the data of a set $G$ with an associative binary operation $\circ : G \times G \to G$, an *identity* element $e$, and the existence of inverse elements. Associativity implies that $f \circ (g \circ h) = (f \circ g) \circ h$ for any $f, g, h \in G$. The identity $e$ is characterized by $e \circ g = g \circ e = g$ for any $g \in G$. Finally, for $g \in G$, the existence of inverses asserts that there is an element $g^{-1} \in G$ such that $g \circ g^{-1} = e$.

A subset $H$ of $G$ is a **subgroup** if it is closed under the group operation $\circ$ and forms a group on its own. Subgroups partition $G$ into equally sized disjoint subsets called **cosets**. A **left coset** is the set $gH = \{gh : h \in H\}$, and similarly, $Hg$ is a **right coset**.

For example $(\mathbf{Z}, +)$, the integers under addition, forms a group, and the even integers form a subgroup. The cyclic group $C_n = \{0, 1, ..., n-1\}$ under addition modulo $n$ is a central object of study in this paper. Subgroups of $C_n$ take the form $\{0, m, 2m, \ldots, (n/m - 1)m\}$ for divisors $m$ of $n$, which corresponds to "wrapping" $C_n$ at a coarser granularity.

A **group homomorphism** is a map $f : G \rightarrow H$ preserving the group operation: $f(g \circ_G h) = f(g) \circ_H f(h)$. If $f$ is bijective, it is called an **isomorphism**.

A **group representation** is a homomorphism $\rho : G \rightarrow \mathrm{GL}(V)$, where $\mathrm{GL}(V)$ is the group of invertible linear transformations on a vector space $V$ over $\mathbf{R}$ or $\mathbf{C}$. Representations link group theory to linear algebra, enabling intuitive analysis via matrices. For example, the cyclic group $C_n$ has **complex representations** $\rho(k) = \exp(2\pi i \frac{mk}{n})$ for $m \mid n$. These encode rotations in the complex plane and correspond to the **discrete Fourier transform (DFT)**. Real representations, such as rotation matrices in $\mathbf{R}^2$, are derived from these complex representations. In this paper, we draw heavily on the connection between representations of $C_n$ and the DFT to analyze how neural networks encode modular addition.

The **Chinese remainder theorem (CRT)** states that if an integer $n$ has factorisation $n = q_1 q_2 \cdots q_j$ into pairwise coprime integers $q_i$, then for any integers $m_1, \ldots, m_j$, the system of congruences $k \equiv m_i \pmod{q_i}$ for $i = 1, \ldots, j$ has a unique solution modulo $n$. In the language of groups, one would say that the map $C_n \rightarrow C_{q_1} \times \ldots \times C_{q_r}$ mapping $k$ to the vector $(k \mod q_1, k \mod q_2, ..., k \mod q_r) = (k_1, ..., k_r)$ is a group isomorphism. One way of finding the solution of this set of congruence relations is to find the preimage of each reduction mod $q_i$ and taking their intersection. This intersection of sets will contain a single element which is the sought solution.

**Remark 1.** *Here is an alternative perspective in a spirit closer to the operations performed by the neural networks we study. Think of each congruence $k \equiv m_i \pmod{q_i}$ as defining a **signal** over $C_n$. To each congruence condition $k \equiv m_i \pmod{q_i}$, associate the function $f_i : C_n \rightarrow \mathbf{R}$ defined by $f_i(k) = \cos(2\pi i \frac{q_i(k - m_i)}{n})$, which evaluates to 1 on the solution set of this congruence. Summing these signals over all congruences, $\sum_i f_i(k)$, produces a function that peaks uniquely on the single solution of the system of modular equivalences.*

**Example: Addition modulo 91.** Consider $n = 91$ with prime factors 7 and 13. Suppose we are solving $(a + b) \mod 91 = 10$. Using CRT and cosets:

Compute $10 \mod 7 = 3$. The set of integers congruent to 3 mod 7 (a coset) is $\{3, \mathbf{10}, 17, 24, ...\}$

Compute $10 \mod 13 = 10$. The set of integers congruent to 10 mod 13 (a coset) is $\{\mathbf{10}, 23, 36, ...\}$

The only number in the intersection of these two sets is $\mathbf{10}$, *i.e.* the unique solution modulo 91. To use this to construct what the neural network is learning when trained on modular arithmetic, we do the following: 1) fit a sine wave through these cosets with a frequency such that it peaks on each value in the coset (in this case that frequency is the prime factor); 2) fit a sine wave through every coset we did not mention, e.g. (mod $3 = 0$, mod $3 = 2$, etc); 3) when we cluster all sine waves (neurons) with the same frequency, we get three clusters and the behavior shown in Figures 6, 18 and 32! This connects the CRT to the neural network's learned representation of modular addition.

**Approximate cosets.** While CRT relies on exact cosets, neural networks often learn more flexible structures that we term approximate cosets. Pick any $c, v \in C_n$ and $1 \leq k \leq n$, then the set $\{c, c+v, c+2v, ..., c+kv\}$ is an **approximate coset.** In fact, we can show that the set of values $a$ with $\cos(2\pi f a/p) > d$ for a fixed value $d$ (*i.e.* exceeds a fixed threshold) forms an approximate coset. To see this, set $v$ to be the modular inverse of $f$, that is, $vf = 1 \mod n$, and pick the first positive integer $c$ such that $\cos(2\pi c/n) > d$ but $\cos(2\pi(c+1)/n) \leq d$. Then we have $\cos(2\pi x/n) > d \iff x \in \{-c, ..., c\}$, which means that $\cos(2\pi f a/n) > d \iff a \in \{-cv, -cv+v, -cv+2v, ..., cv\}$. If you introduce a phase shift to the cosine, then that just changes the starting point of the approximate coset. Section 5.1 goes into more detail for the case when the frequency does not divide $n$, as will be the case most of the time, and shown in Fig 3 .

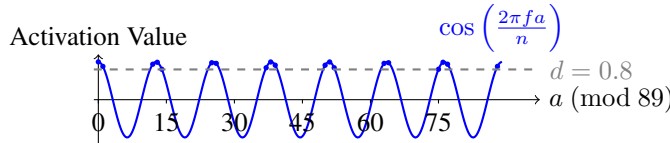

Figure 3: Visualization of approximate cosets. The sinusoidal function $\cos\left((2\pi f a)/n\right)$ is plotted over the modular values $a \in C_n$ for $n = 89$ and frequency $f = 7$. Points where the cosine value exceeds the threshold $d = 0.8$ are highlighted in blue, forming the approximate coset. The dashed line represents the threshold level. This figure illustrates how neurons encode modular arithmetic using thresholded sinusoidal activations.

| Study | Embeddings | Neurons |
|---|---|---|
| Nanda et al. (2023a) | Rotation matrices | "Clock" |
| Chughtai et al. (2023b) | Representation matrices | Matrix mult. |
| Zhong et al. (2024) | Circular, Lissajous curves | "Pizza" |

Table 1: Different interpretations of neural network components across studies.

## 3.2 TASK DESCRIPTION AND MODEL

The task is to learn the labeling function $\circ : C_n \times C_n \to C_n$, corresponding to addition modulo $n$ in the cyclic group $C_n$. The dataset is the $n^2$ pairs $(a, b)$ with label $c = a + b \mod n$. We train 128-neuron (with biases) 1-hidden-layer ReLU networks with two embedding matrices (dimension 32), one for $a$ and one for $b$, with $n$ output logits. The training set is 5096 randomly selected pairs for $n = 91$ and 4717 pairs for $n = 89$. The inputs $a$ and $b$ to the neural network select a column from their respective embedding matrix, then these columns are concatenated, and the neural network receives this concatenated vector as input. We train models using the Adam optimizer (Kingma & Ba, 2015), with a learning rate of $0.008$ and L2 weight decay of $0.0001$ (Note we use $\|\boldsymbol{\theta}\|_2^2$, not the square root, where $\boldsymbol{\theta}$ is the weight vector of the network).

## 4 INVESTIGATING NETWORKS TRAINED ON MODULAR ADDITION

Diverging observations have been made across works. Despite these differences, a closer examination reveals that these networks exhibit invariances at three key points: **embeddings, neuron pre-activations, and logits**. These invariances suggest a consistent high-level structure across models, even if the specific mechanisms vary. See Table 1 for a summary of the various interpretations of networks across the literature at these key points.

While these invariances provide a unifying thread, the lack of a cohesive framework to explain how they arise has led to confusion and conflicting interpretations. In the next section, we present experimental findings that help us uncover these invariances and provide the foundation for our unified CRT-inspired model.

### 4.1 OUR EXPERIMENTAL FINDINGS

Here we present evidence implying that since the network has the same distribution of learned frequencies for both composite and prime moduli, it is likely to be using the same algorithm for both cases. We also discuss the existence of fine-tuning neurons.

**Inspecting the preactivations: simple and fine-tuning neurons.** We split the preactivation for a neuron into two parts: one coming from the column of the embedding matrix for $a$ and the other from $b$. This reproduces past findings in the literature that the preactivations are periodic functions (Pearce et al., 2023). Furthermore, by taking the discrete Fourier transform (DFT) of the preactivations, we find that most neurons have periodic functions that concentrate on a single frequency and its negation mod $n$ (and thus on a single complex representation of $C_n$ and its conjugate). We call these neurons "simple" neurons, but we discover an additional type of neuron which concentrates

on linear combinations of complex representations, made evident by 2 & Fig. 22 in B.1.2. This type of neuron does not show up in every random seed and thus, it is not necessary for it to be learned. This is part of the reason that we call these neurons "fine-tuning neurons", see Fig 2 to see the difference between a simple and fine-tuning neuron. Note that the sum of two periodic functions is another with the same frequency, so the postactivations are also periodic, but with the negative part clipped to 0 by ReLU. We also perform an analysis of the phases that the periodic functions learned by neurons are shifted by *e.g.* see Fig. 25a and Fig. 25b, which are relevant to section A.2.

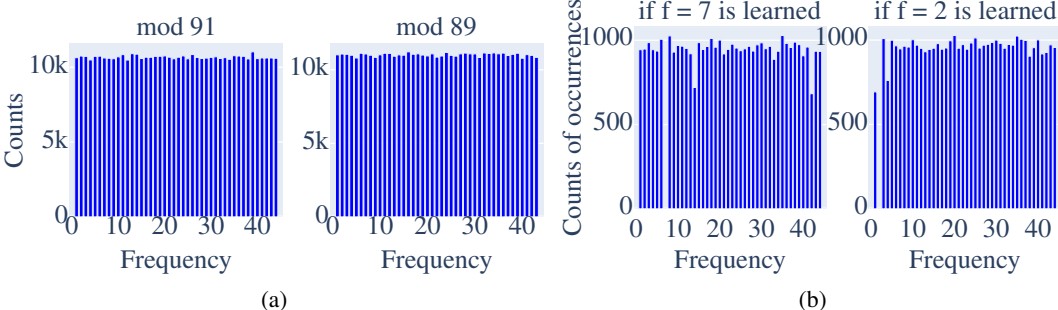

Figure 4: a) Histograms of frequencies found across 100k random seeds for mod 91 (factors 7 and 13) and mod 89 are both uniform. b) The conditional histograms of frequencies over 100k seeds, both mod 91. Left: if frequency 7 is found then neurons with $f = 14$ or $f = 43$ are less likely (note that $2 \cdot 43 \equiv -7 \pmod{91}$). Right: if frequency 2 is present then frequencies 1 and 4 are less likely.

**Investigating distribution of learned frequencies.** Since previous work only studied prime moduli, and given Stander et al. (2023) discovered coset structure in the neurons: here our aim is to determine whether neural networks learn coset structure for modular addition. For composite $n$, this would mean finding frequencies that exactly divide $n$. For prime $n$, we would be looking for an approximate coset structure since $n$ has no divisors. First, by training 100,000 neural networks on $n = 89$ and $n = 91$ we show that the distribution of learned frequencies is uniform, see Fig 4a or Fig. 24. The uniformity across frequencies coprime to $n$ is due to the existence of group isomorphisms: if $f$ is coprime to $n$, then replacing each $x$ with $fx \mod n$ does not change the group structure, and so the neural network cannot tell the difference between the two. This observation allows us to examine neurons with different frequencies from a more uniform perspective. If the periodic function has frequency $f$, we will transform the $x$-values by multiplying them by $\frac{f}{\gcd(f,n)}x \mod n$. See Fig 2 for a visualization of this transformation applied to a simple neuron and a fine-tuning neuron remapped by this process with $n = 89$.

Note, this process results in every neuron being remapped to a function with frequency $\gcd(f, n)$, which is 1 for most values of $f$. The preactivations for simple neurons are transformed by this process into a standard sine wave with a phase shift, and those for fine-tuning neurons are transformed into one half the period of a sine wave. This provides a "standard" reference frame for ease of analysis of individual neurons of a given frequency, and allows us to display the difference between simple and fine-tuning neurons to better understand their behaviour. Note that we cannot apply the same group isomorphism to all neurons at the same time to standardize their frequencies in this way; rather, each collection of neurons with the same frequency must be considered separately.

**Investigating conditional distribution of frequencies.** While the distribution of frequencies is uniform on the whole, if we condition on any given frequency, the distribution of other frequencies the network learns fails to be uniform (see Fig. 4b). In particular, we observe that if a network learns $f$, it is less likely to learn $2f$ and $\frac{f}{2}$ modulo $n$. This supports the interpretation that the network is learning an approximate CRT algorithm, due to frequencies with $2f$ and $\frac{f}{2}$ making it harder to lower the loss if $f$ is learned. This follows because they intersect, taking large values on the same logits that are not the correct logit.

**Clustering neurons.** We cluster neurons based on their preactivations having the same frequency. We inspect the contribution of these clusters to the output logits of the neural network, cluster by cluster, see Fig. 6 and notice that the clusters shift their phase and magnitude based on $(a, b)$. We do this by decomposing the dot product into the linear combinations coming from neurons in

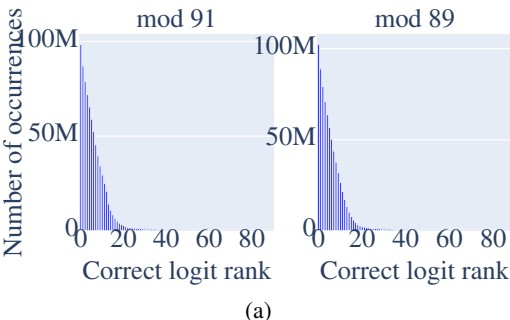 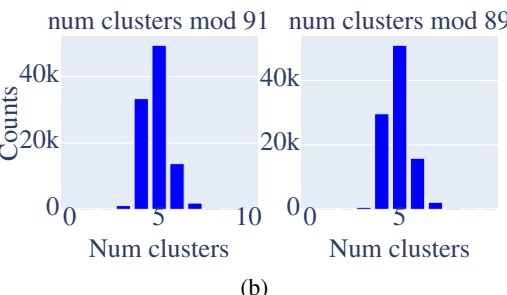

Figure 5: a) Histograms of the correct logit rank if only a single cluster was trusted to output the correct logit, *i.e.* if argmax of any one cluster was used as the output. b) The histogram of the number of clusters found across 100k random seeds for $n = 91$ and $n = 89$.

a cluster to each logit. This is essentially the same process as previously explained for isolating the preactivations from just embedding matrices $a$ or just $b$. Finally, we assess the distribution of logit correctness within clusters by examining the rank of the correct logit when logits are sorted by decreasing magnitude. If the correct logit ranks 0, it implies that using the argmax over that cluster would yield the correct result. Figure 5a shows the correctness histogram over 20,000 training runs.

# 5 BRIDGING PERSPECTIVES FOR MODULAR ADDITION

In this section we will attempt to unify the literature and explain that the different conclusions by different authors on this problem aren't so different. By explicitly modeling single neurons similar to Gromov (2023), we will explain why representations (Nanda et al., 2023a), (Chughtai et al., 2023b); cosets (Stander et al., 2023) and Lissajous-like curves Zhong et al. (2024) are seen by other papers. Our interpretation applies to cyclic groups of both prime and composite order and is robust to different batch sizes.

**A model for simple neurons.** Suppose we are trying to learn addition mod $n$. Label the input embedding columns as $A_0, \ldots, A_{n-1}, B_0, \ldots, B_{n-1}$, and the output logits as $D_0, \ldots, D_{n-1}$. Let $w(U, V)$ be the dot product of all values from $U$ with their edge weights leading to $V$. Here is a model for a "simple neuron" $N$, with parameters $f, s_A, s_B \in C_n$ and positive real number $\alpha$ such that for each $k \in C_n$ we have

$$w(A_k, N) = \cos \frac{2\pi f(k-s_A)}{n}, \quad w(B_k, N) = \cos \frac{2\pi f(k-s_B)}{n}, \quad w(N, D_k) = \alpha \cos \frac{2\pi f(k-s_A-s_B)}{n}.$$

We find experimentally that many neurons match this model quite closely. See Section A.1 for an explanation of how to detect simple neurons. This model of simple neurons will be used in the following section to show how the neuron's activations are added to compute the correct output.

## 5.1 SIMPLE NEURONS LEARN COSETS

A neuron satisfying the above computes a trigonometric function that has its maxima on the elements of a coset or "approximate coset". If $g := \gcd(f, n) > 1$, the neuron has learned the **coset** of order $g$ containing $s_A + s_B$. More precisely: writing $n = n'g$ and $f = f'g$ for $g = \gcd(f, n)$, we can rewrite $\frac{2\pi f}{n} = \frac{2\pi f'}{n'}$. So if the input neurons are at positions $a$ and $b$ where $a \equiv s_A \pmod{n'}$ and $b \equiv s_B \pmod{n'}$, then the activation of the neuron has a maximum: $\cos \frac{2\pi f(a-s_A)}{n} = \cos \frac{2\pi f(b-s_B)}{n} = 1$. The neuron points most strongly to every logit satisfying $c \equiv s_A + s_B \pmod{n'}$, because for all such output logits $\cos \frac{2\pi f(c-s_A-s_B)}{n} = 1$. We see that the neuron strongly associates elements of $C_n$ that are congruent modulo $n'$.

Whether $f$ is a divisor of the order $n$ or not, the neuron will activate on what we defined as an approximate coset. More precisely, we can ask the following: for $a \not\equiv s_A \pmod{n'}$, which values of $a$ have the largest activation? We have $\cos \frac{2\pi f'(a-s_A)}{n'}$ very close to 1 if and only if $f'(a - s_A)$ is very close to an integer multiple of $n'$; that is, say, $f'(a - s_A) \equiv m \pmod{n'}$ for some integer

$m$ with small absolute value. Letting $d$ denote a modular inverse of $f'$ mod $n$, this is equivalent to $a - s_A \equiv dm \pmod{n'}$. In other words, by taking $a = s_A + dm$ for small integers $m$, the neuron will be activated very strongly. Likewise if $b = s_B + dm'$ for some other small integer $m'$. Now this neuron will point most strongly to $c \equiv s_A + s_B \pmod{n'}$ as discussed above, but if $c$ is a small number of steps of size $d$ away from $s_A + s_B$, it will still have large activation. To summarize: if you can reach each of $a, b, c$ via a small number of steps of size $d$ from $s_A, s_B, s_A + s_B$, respectively, then $N$ fires strongly on inputs $a, b$, and points strongly at $c$.

Following 1, we can now present an idealised version of the further computations of the neural network that finally compute the group operation as output. Assume that the network has neurons that have learned frequencies $f_1, \ldots, f_r$ as described above. Adding the activations of each of those neurons will superimpose signals that will interfere constructively at the correct answer and destructively at all other answers; that is, the correct answer is identified as a unique element in an intersection of approximate cosets. If the frequencies are relatively coprime and $n = f_1 \cdots f_r$, then the global maximum of the output function recovers a unique solution to a system of modular congruences, essentially recovering the Chinese remainder theorem; see Remark 1. In general, if $f_1, \ldots, f_r$ are not divisors of $n$, this superposition process is described in more depth in Section 5.2.

## 5.2 A MODEL FOR WHAT THE NETWORK LEARNS — THE APPROXIMATE CRT ALGORITHM

In this section we provide a mathematical model that matches experimental results acquired under typical machine learning conditions, *i.e.* the dataset is split into a training set and test set, and the neural network is trained with hyperparameters found after hyperparameter tuning. This argument is not a rigorous proof. This heuristic better aligns with our empirical results and those in the literature; results suggesting networks learn closer to $\mathcal{O}(\log(n))$ frequencies than $\frac{n-1}{2}$.

Each simple neuron maximally activates a single output, namely $s_A + s_B$ (or possibly a coset containing $s_A + s_B$). However, if we combine the contributions from all simple neurons in a single cluster (i.e. all with the same frequency $f$), we observe that the activation level can be maximized at any desired output; more precisely, the activation level at output $k$ given inputs $i, j$ will be of the form $A \cos(2\pi(f(k - i - j)/n))$. Note this has been observed experimentally *e.g.* see Fig 25a and Fig. 6, and has also been previously noticed in the literature: see e.g. the last equation of section 3 in Chughtai et al. (2023a). In fact, the analysis below still works even if $i + j$ is only somewhat close to the maximal activation of the cluster (see 6), though we assume the maximum is at $i + j$ for simplicity. However, even in this case there will be many output neurons that all activate nearly as strongly as the correct answer. To isolate a single answer, we use a superposition of sine waves of multiple frequencies; we observe experimentally in Fig 6 that this process makes the correct answer stand out from the rest.

In light of the above, if we fix input neurons $i, j \in C_n$ then combining the contributions from all clusters (and assuming for the heuristic that the contributions from all clusters have the same amplitude), the sum $h(k) := \sum_{\ell=1}^{m} \cos\left(\frac{2\pi f_\ell}{n}(k - i - j)\right)$ gives a model for the activation energy at output neuron $k$. If $k = i + j$, then $h(k)$ takes on the maximum value $m$. If we want to guarantee the neural net will consistently select $k$, we need to show that $h(k)$ is significantly less than $m$ for all other values of $k$. We'll assume a random model where $m$ frequencies $f_1, \ldots, f_m$ are chosen uniformly at random from $1, 2, \ldots, n - 1$. Fix a parameter $0 < \delta < 1$; we will compute an approximation for the probability that $m - h(k) > \delta m$ for all $k \neq i + j \pmod{m}$.

Let $\{x\} := x - \lfloor x + \frac{1}{2} \rfloor$ be the signed distance to the nearest integer and set $d := k - i - j$. Then using a Taylor expansion,

$$m - h(k) = m - \sum_{\ell=1}^{m} \cos\left(2\pi\left\{\frac{f_\ell d}{n}\right\}\right) \approx m - \sum_{\ell=1}^{m}\left(1 - \frac{1}{2}\left(2\pi\left\{\frac{f_\ell d}{n}\right\}\right)^2\right) = 2\pi^2 \sum_{\ell=1}^{m}\left\{\frac{f_\ell d}{n}\right\}^2.$$

Note that the Taylor approximation is quite bad when $f_\ell d/n$ is far from an integer, and if $k$ is close to an integer then $\{k\}$ is close to 0. It is reasonable to expect that $m - h(k)$ will be minimized when the values $f_\ell d/n$ are all close to integers, in which case the approximation is more accurate.

Thus the condition $m - h(k) > \delta m$ is related to the following condition: defining the vector $v := \frac{1}{n}(f_1, \ldots, f_m) \in [0,1]^m$, we need that for all $1 \leq d \leq n - 1$, the point $dv$ has distance at least $\sqrt{\delta m/2\pi^2}$ away from any point in $\mathbf{Z}^m$. Note that $nv$ is an integer point, so $(n-d)v$ is always the

same distance from an integer point as $dv$ is. Thus it suffices to require $v$ to be at least $\frac{1}{d}\sqrt{\delta m/2\pi^2}$ away from a point in $\frac{1}{d}\mathbf{Z}^m$ for $d = 1, \ldots, \lfloor n/2 \rfloor$.

We compute an upper bound on the volume of the region to be avoided: that is, the set of all points in $[0,1]^m$ within $\frac{1}{d}\sqrt{\delta m/2\pi^2}$ of a point of $\frac{1}{d}\mathbf{Z}^m$ for some $d = 1, \ldots, n/2$. For each $d$, there are $d^m$ points in this region, and each has a ball of radius $\frac{1}{d}\sqrt{\delta m/2\pi^2}$ around it; the total volume of the region to be avoided is therefore bounded above by $\frac{n}{2\Gamma(m/2+1)}\left(\frac{\delta m}{2\pi}\right)^{m/2}$. Thus the probability that $m - h(k) > \delta m$ is approximately equal to 1 minus this value.

For a given $n$, let's compute the value of $m$ that makes this probability greater than, say, $\rho$.

$$1 - \frac{n}{2\Gamma(m/2+1)}\left(\frac{\delta m}{2\pi}\right)^{m/2} > \rho \iff \Gamma(m/2+1)\left(\frac{2\pi}{\delta m}\right)^{m/2} > \frac{n}{2 - 2\rho}.$$

Taking a natural logarithm, applying Stirling's approximation $\log_e \Gamma(x+1) \approx x\log_e(x) - x$, and solving for $m$,

$$\log_e \Gamma(m/2+1) + \frac{m}{2}\log_e\left(\frac{2\pi}{\delta m}\right) > \log_e n - \log_e(2 - 2\rho)$$

$$m > \frac{2\log_e n - 2\log_e(2 - 2\rho)}{\log_e(\pi/\delta) - 1}.$$

Thus if the number of neuron clusters $m$ is greater than this expression, then with probability at least $\rho$, the separation $m - h(k)$ will be at least $\delta m$. We see that the number grows linearly in $\log_e n$.

Choosing the parameters $\rho$ and $\delta$ can significantly change the precise value of $m$ needed, and it's not clear which values most accurately model the true behavior of the neural net. As an example, note that if we take $\delta = \pi/e^3 \approx 0.1564$, and $\rho = \frac{1}{2}$, then this whole expression simplifies to just $m > \log_e n$. Thus, if the neural net uses $m = \log_e n$ neuron clusters, then this heuristic predicts that it will guarantee a separation $m - h(k) > 0.15m$ for all $k \neq i + j$ with 50% certainty. For $n = 89, 91$ we have $\log_e n \approx 4.5$, which agrees with the number of clusters found in Figures 5b, 30 and 31. This process can be interpreted as an "approximate CRT;" see Remark 1 for the analogy.

### 5.3 Connecting our experimental results with our model

Our experimental findings show that the neurons concentrate on the complex representations of the group thus learning periodic functions; see Fig. 2. Simple neurons concentrate on one complex representation and fine-tuning neurons concentrate on multiple complex representations (Figs 2, 22). Additionally, fine-tuning neurons have additive or subtractive relations with each other, see Fig. 29, while simple neurons try to avoid these relations Fig. 4b. Furthermore, we introduced a model for a simple neuron that explains that the simple neurons are actually activating on (approximate) cosets. The best way to see this is to relate it to the CRT which uses set intersection on cosets (see 3.1), whereas the network superimposes linear combinations of cosets onto the logits by using different frequencies, thus giving the correct logit the highest value (guaranteeing its selection by argmax). Furthermore, we show how using cosets in this way models the constructive and deconstructive interference, explaining why the incorrect logits have low values instead of just why the correct logit has a high value, see Fig. 6 and Section 5.2. We'd also like to point out that the pizza interpretation is an implementation of the CRT see Fig. 18, *e.g.* see Fig. 1 in Zhong et al. (2024), "Same-label predictions are spread over two slices of pizza" proceeds to list all points in the coset $a + b \equiv 5 \mod 12$, and see Section B.1.1.

Furthermore, the circular, non-circular and Lissajous-like curves found in Zhong et al. (2024) are predicted by our simple neuron model. Given two neurons in different clusters, *i.e.* with different frequencies $(f_1, f_2)$, a parametric plot of their activations $(\cos(2\pi f_1 t), \cos(2\pi f_2 t))$ traces out a Lissajous curve, *i.e.* non-circular embedding. We show that the non-circular embeddings in Zhong et al. (2024) result from principal components belonging to different clusters (cosets) in B.1.

Conjecture 1 is implied by our heuristic model in 5.2 and scaling experiments in Figures 30 and 31.

**Conjecture 1:** *when training with biases, cross entropy loss and $L_2$ weight norm as a regularization penalty, a good local minima for learning $a + b \mod n$ results in the neural network learning $\mathcal{O}(\log(n))$ clusters of different frequencies in order to minimize the loss sufficiently.*

# 6 CONCLUSION AND DISCUSSION

We have unified experimental findings with a model for an approximate CRT, demonstrating that the following are different expressions of the same overall phenomenon: each neuron learns a projection of a representation, i.e., a phase-shifted representation, or an approximate coset of a specific subgroup (learning a coset if the frequency is a prime factor). Furthermore, we demonstrate that superimposing neurons to form clusters reveals that neurons come together to construct cosets that are approximately equivariant to changes to the inputs of the network independently of whether the network is a clock Figs. 6 and 32 or pizza 18. Thus, we show that the algorithm the model is using to minimize the loss is reminiscent of the CRT—using linear combinations of cosets. Thus, it is the case that the results found by Nanda et al. (2023a); Chughtai et al. (2023b); Zhong et al. (2024) are all true, yet simultaneously conflict with eachother. This is because the explanations are not robust at multiple scales, giving rise to an interesting question: should we consider an interpretation to be good only if it's true at more than one scale of abstraction?

Furthermore, the result of Stander et al. (2023) that networks trained on permutation groups are learning cosets and not representations, is no longer a conflicting piece of evidence for a universality hypothesis that neural networks learn similiar structures when trained on similar classes of data (finite groups). We restore hope for the universality hypothesis to be true by showing that cosets – not GCR – are core features in networks learning the cyclic group across a variety of hyperparameter conditions. Our work revisits the conjecture, which was believed to be refuted, and shows that both solutions can be unified, reopening the conjecture for consideration.

**Reopened-conjecture 2:** *There is universality in the structures neural networks uncover when trained on group operations. This universality involves coset circuits and approximate cosets.*

It's worth noting that researchers working on new approaches to a theory for deep learning have been in search of a model that appears to have learned features of an error-correcting code (Murfet, 2024). The aforementioned superpositions of approximate cosets is where these features are contained. We can see in Fig. 5a that the ability of a single cluster to output the correct answer varies a lot. The reason the neural network learns different frequencies is a way of encoding redundant information to ensure a large separation between the ultimate logit output and the second largest output. This is reinforced by Fig. 27 finding that all weights in a cluster can have a substantial amount of multiplicative noise injected, without destroying test accuracy.

Future work should address Conjecture 1 as our $\mathcal{O}(log(n))$ model gives hints about how superposition is behind the unreasonable effectiveness of neural networks. A model embracing it achieved results matching all experiments in the literature. Future interpretability research should focus on Conjecture 2; especially our observation leading to it: that the only interpretation that unifies all interpretations is true at multiple scales. If conjecture 2 is true it will aid attempts in automating circuit discovery in trained networks and benefit AI safety research.

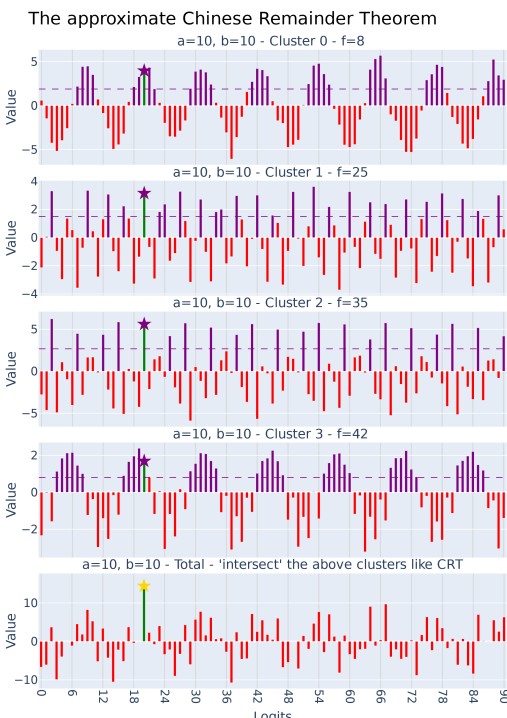

Figure 6: **The approximate CRT visualized:** Four clusters and their contributions to each output logit. The fifth row of plots is the final logit values. The correct logit is in green with a star. This random seed has $f = \{35, 25, 8, 42\}$. Purple bars are in the approximate coset of the cluster.

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

# A APPENDIX

## A.1 DETECTING SIMPLE NEURONS

Recall that a simple neuron comes with parameters $f, s_A, s_B \in C_n$ and positive real number $\alpha$ such that for each $k \in C_n$ we have

$$w(A_k, N) = \cos \frac{2\pi f(k - s_A)}{n}, \quad w(B_k, N) = \cos \frac{2\pi f(k - s_B)}{n}, \quad w(N, D_k) = \alpha \cos \frac{2\pi f(k - s_A - s_B)}{n}.$$

We verify that a neuron satisfies this model by using the fact that any function $h : \mathbf{Z}/n\mathbf{Z} \to \mathbf{C}$ is uniquely determined by its discrete Fourier transform (DFT) $\widehat{h}(\omega) = \sum_{k=0}^{n-1} h(k)e^{-2\pi i \omega k/n}$, which then satisfies $h(k) = \frac{1}{n}\sum_{\omega=0}^{n-1} \widehat{h}(\omega)e^{2\pi i \omega k/n}$. Suppose we're given a neuron $N$. We can compute all dot products joining $N$ to input and output neurons: $w(A_k, N)$, $w(B_k, N)$, and $w(N, D_k)$ for all $k = 0, 1, \ldots, n-1$. We then compute the DFTs of these three functions. In order to be a simple neuron, there must exist a single value $f$ such that all three DFTs have large values at $f$ and $-f$, and negligible values elsewhere. We assume this condition going forward.

Consider $h(k) = w(A_k, N)$. Since by assumption $\widehat{h}(\omega)$ is close to zero except at $f$ and $-f$, we have $h(k) = \widehat{h}(f)e^{2\pi i f k/n} + \widehat{h}(-f)e^{-2\pi i f k/n}$. Note that the complex conjugate of this expression is

$$\overline{h(k)} = \overline{\widehat{h}(-f)}e^{2\pi i f k/n} + \overline{\widehat{h}(f)}e^{-2\pi i f k/n}.$$

Since all weights are real numbers, we have $\overline{h(k)} = h(k)$, so by uniqueness of DFT we must have $\widehat{h}(-f) = \overline{\widehat{h}(f)}$. Writing $\widehat{h}(f)$ in polar form $re^{i\theta}$, and setting $t := -\frac{n\theta}{2\pi}$, we have

$$h(k) = re^{i\theta}e^{2\pi i f k/n} + r_A e^{-i\theta}e^{-2\pi i f k/n}$$

$$= re^{2\pi i(fk-t)/n} + re^{-2\pi i(fk-t)/n}$$

$$= 2r \cos \frac{2\pi(fk-t)}{n}.$$

A similar argument holds for $h(k) = w(B_k, N)$ and $h(k) = w(N, D_k)$, so for some parameters $r_A, r_B, r_D, t_A, t_B, t_D$ we have $w(A_k, N) = 2r_A \cos \frac{2\pi(fk-t_A)}{n}$, $w(B_k, N) = 2r_B \cos \frac{2\pi(fk-t_B)}{n}$, $w(N, D_k) = 2r_C \cos \frac{2\pi(fk-t_D)}{n}$. Now check experimentally that $r_A = r_B$ and $t_A + t_B = t_D$ (up to small error). Note that we can rescale all the weights connected to $N$ without changing how the neural net operates: dividing all input weights by some positive constant $\tau$, and multiplying the output weights by $\tau$, will not change the contributions to each output logit from that neuron. So rescaling by $2r_A$ we have

$$w(A_k, N) = \cos \frac{2\pi(fk-t_A)}{n}, w(B_k, N) = \cos \frac{2\pi(fk-t_B)}{n}, w(N, D_k) = \alpha \cos \frac{2\pi(fk-t_A-t_B)}{n}$$

for some positive $\alpha$.

Now round $t_A$ to the nearest integer multiple of $g := \gcd(f, n)$, say $t_A \approx gm_A$. By Bezout's identity we can write $gm_A = fs_A + nu_A$ for some integers $s_A, u_A$, so we have

$$w(A_k, N) \approx \cos \frac{2\pi(fk - fs_A - nu_A)}{n} = \cos \frac{2\pi f(k - s_A)}{n}, \text{ and}$$

$$w(B_k, N) \approx \cos \frac{2\pi f(k - s_B)}{n}, \quad w(N, D_k) \approx \cos \frac{2\pi f(k - s_A - s_B)}{n},$$

showing that this neuron fits the model of a simple neuron.

Summarizing: to check that a neuron satisfies the simple model:

1. Compute the DFTs of $h_A(k) := w(A_k, N)$, $h_B(k) := w(B_k, N)$, and $h_D(k) := w(N, D_k)$. Check that for some $f$, we have $\widehat{h_A}(\omega) \approx \widehat{h_B}(\omega) \approx \widehat{h_D}(\omega) \approx 0$ for all $\omega \neq f, -f$.

2. Write the values of the DFT at $f$ as
$$\widehat{h_A}(f) = r_A e^{-2\pi i t_A/n}, \qquad \widehat{h_B}(f) = r_B e^{-2\pi i t_B/n}, \qquad \widehat{h_D}(f) = r_D e^{-2\pi i t_D/n}.$$
Check that $t_A + t_B \approx t_D$ and $r_A \approx r_B$.

If the above tests both hold, then we can replace $N$ with a simple neuron without drastically changing the functioning of the neural net.

## A.2 EMBEDDINGS CONTAIN PROJECTIONS OF REPRESENTATIONS, NOT REPRESENTATIONS

Chughtai et al. (2023b) discover representation values in the embedding matrix. The first step in their GCR algorithm is not true in general. They state "Translates one-hot $a$, $b$ to representation matrices". This is disproven by training with a mini-batch size equal to the modulus $n$ and training with a full batch size. See the difference in the distribution of the resulting embedding matrices in Fig. 7. Furthermore, neurons in a cluster of frequency $f$ have different phase shifts, and $2 \times 2$ rotation matrices in the embeddings doesn't suffice to explain this behaviour.

Instead, the values found in the embedding matrix may encode scaled projections of a $2 \times 2$ rotation matrix onto a one dimensional subspace. Note that such structure is implied by the hypothesis that neural networks trained on group tasks learn representations, but is more general because of the existence of both amplitude and phase shifts. To get an exact equivalence, we note that this neuron structure can be obtained by an *arbitrary scaled projection* of representations. Suppose

$$\rho(k) = \begin{pmatrix} \cos(2\pi f k/n) & -\sin(2\pi f k/n) \\ \sin(2\pi f k/n) & \cos(2\pi f k/n) \end{pmatrix}$$

is a $2 \times 2$ matrix representation of $C_n$. If we apply $\rho(k)$ to the vector $(1, 0)$ and then take the dot product with $(\alpha\cos(2\pi f s_a/n), -\alpha\sin(2\pi f s_a/n))$ (which is the same as projecting onto the subspace spanned by this vector and scaling by $\alpha$) we obtain exactly

$$\alpha\cos\frac{2\pi f k}{n}\cos\frac{2\pi f s_A}{n} + \alpha\sin\frac{2\pi f k}{n}\sin\frac{2\pi f s_A}{n} = \alpha\cos\frac{2\pi f(k-s_A)}{n} = \alpha w(A_k, N).$$

Thus we have explained the phase shifts of different neurons in a cluster, and shown that it's not just the components of $\rho(k)$ that appear in the embeddings, but rather scaled projections of the representations onto arbitrary 1-d subspaces. In our model of simple neurons we ignore the amplitude to make the analysis simpler, but in general it does need to be included. See Fig 2 for example where the amplitudes are greater than 2.

**Inspecting the distribution of embedding matrix weights.** Contrary to findings by Nanda et al. (2023a); Chughtai et al. (2023b), we did not observe the $2\times2$ representation matrix values (used to encode rotations) in our embedding matrices outside their reported training conditions. As shown in Fig. 7, the distribution of embedding weights varies significantly between small and full batch size and the tails of the distributions are quite different. In the case of small batch size, numbers can be found in the range (-2, 2), whereas large batch size contains numbers between (-1.5, 1.5). Note that we choose to remove weights that are between (-0.025, 0.025) to make it easier to see the tails of the distribution; this was done due to 2.4million weights occurring within this range when training with the small batch size. Specifically, in the small batch size regime, around 5% of the weights fell outside the interval $[-1, 1]$, including some weights larger than 2. These values are not consistent with rotation matrix entries. Other than this, we could not identify any significant differences in the core structures of what the neural net learns between the batch sizes.

Combining these experimental findings (Fig. 7) with this model (see A.2) explains that the embedding matrices may contain scaled projections of representations. This explains the different shifts in the periodic functions that can be seen in Figs 25a, 25b and 23, which GCR (Chughtai et al., 2023b) fails to explain.

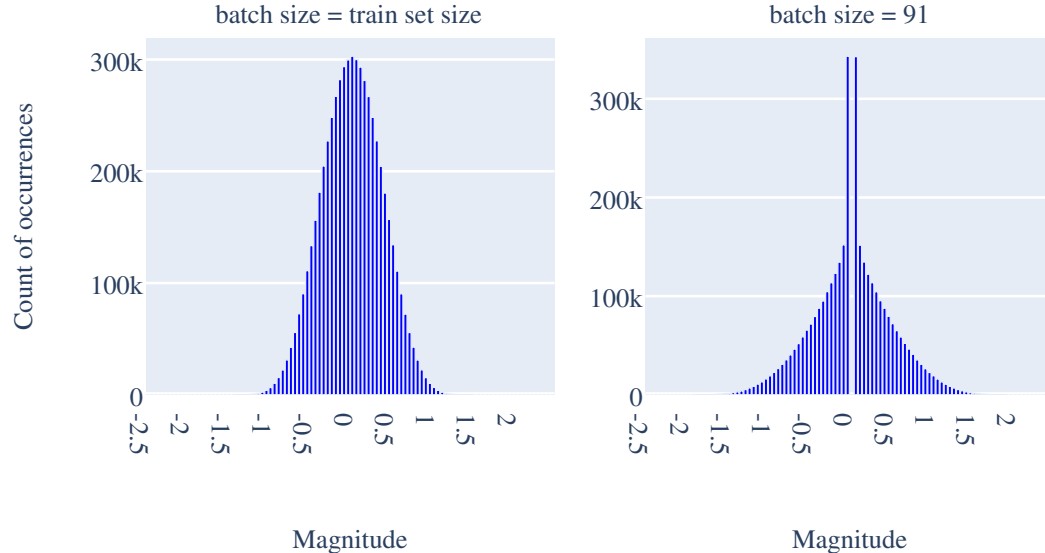

Figure 7: The histograms of embedding weight magnitudes found across 10k random seeds for mod 91 provide evidence against rotation matrices. With batch-size 91 about 5% of the weights are > 1 or < -1, whereas when the batch size is the training set size fewer than 0.5% of the weights are > 1 or < -1. The bin with 0 was removed for batch size 91 due to so many dead weights obfuscating the plot. The value was 2.4 mil, implying that small batches find sparse embeddings with larger magnitude weights.

## B    MORE EXPERIMENTAL EVIDENCE

### B.1    PRINCIPAL COMPONENT ANALYSES OF THE CONCATENATED EMBEDDING MATRIX

We replicate the results of Zhong et al. (2024) and add an additional Fourier transform plot next to their PCA plots, which makes it obvious that the principal components map directly to one cosets with some frequency. It can be seen that all non-circular embeddings and Lissajous embeddings are caused by the two principal components coming from different cosets, as claimed in section 5.3. To make this easy to understand, please see Fig. 8, showing this random seed has four clusters, with key frequencies 35, 25, 8, 42.

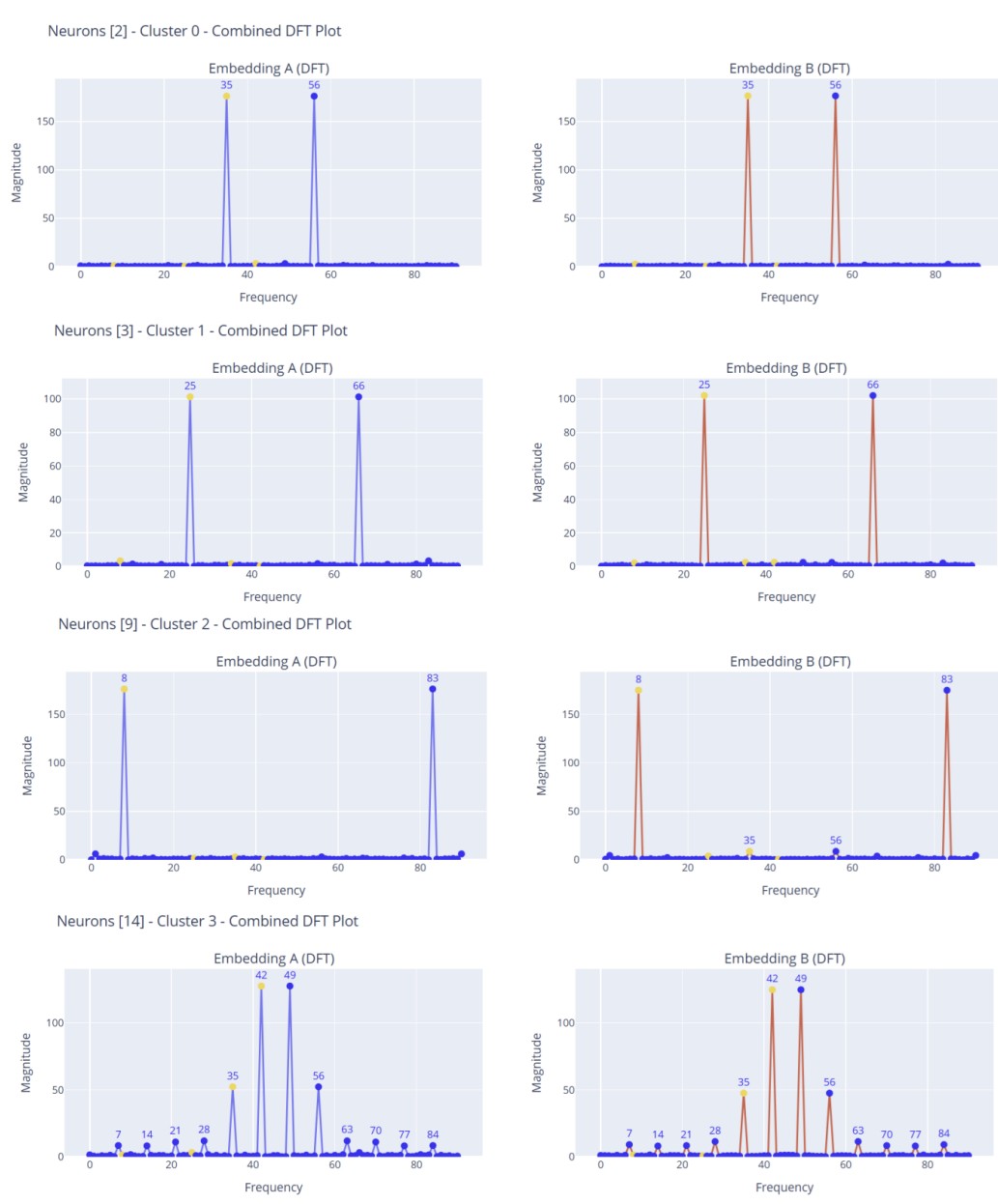

Figure 8: DFT's of neurons in each of the four clusters in this random seed. Cluster 0 has frequency 35, cluster 1 has frequency 25, cluster 2 has frequency 8 and cluster 3 is a fine tuning cluster with frequencies on multiples of 7, 14, 21, 28, 35, 42.

Now below see replications of Zhong et al. (2024), with added DFT plots to support section 5.3.

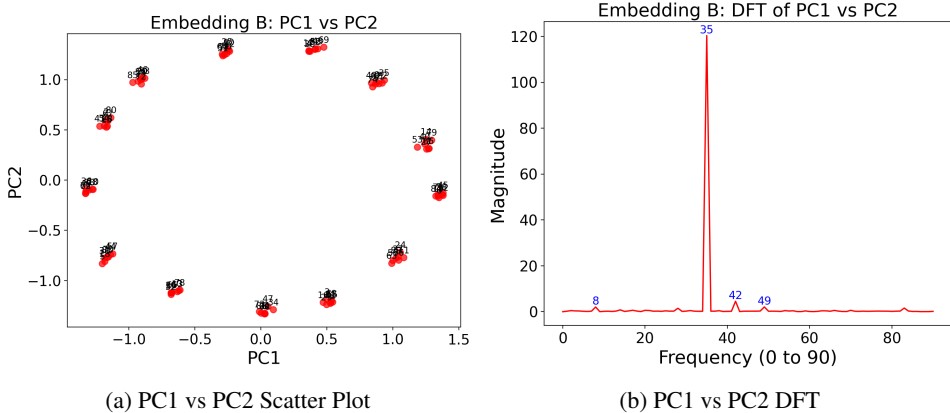

(a) PC1 vs PC2 Scatter Plot

(b) PC1 vs PC2 DFT

Figure 9: PCA and DFT for PC1 vs PC2 showing a circular embedding clustered into cosets. The x and y axis of the left plot are the PC1 and PC2 values for the concatenated embedding matrix for each point $(a, b) \mod 91 \in (0, 0), (1, 1), ..., (90, 90)$. Note that this covers all output classes of the neural network exactly once. Also note that the embedding here is showing 13 cosets with 7 points in them each, *i.e.* all 13 cosets $(a + b) \mod 13 = i, i \in \{0, ..., 12\}$ are in the plot. Both PC1 and PC2 have $f = 35$ and since $gcd(35, 91) = 7$, a prime factor, it's possible to learn the exact cosets.

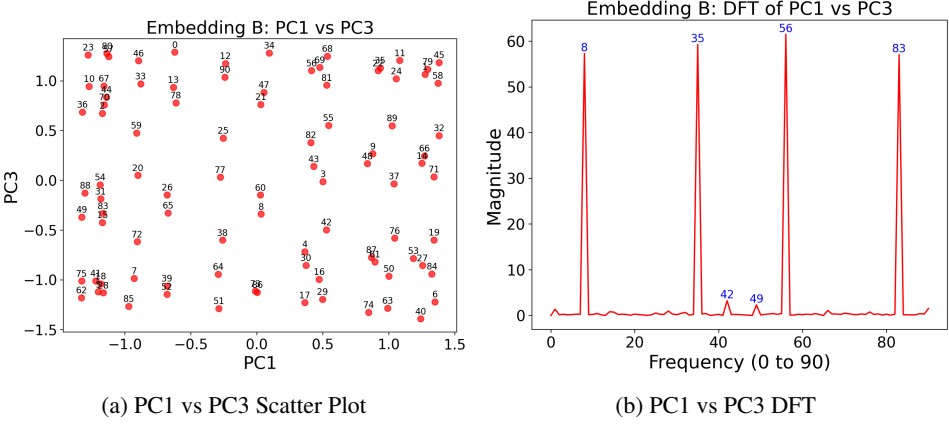

(a) PC1 vs PC3 Scatter Plot

(b) PC1 vs PC3 DFT

Figure 10: PCA and DFT for PC1 ($f = 35$) vs PC3 ($f = 8$), a non-circular embedding.

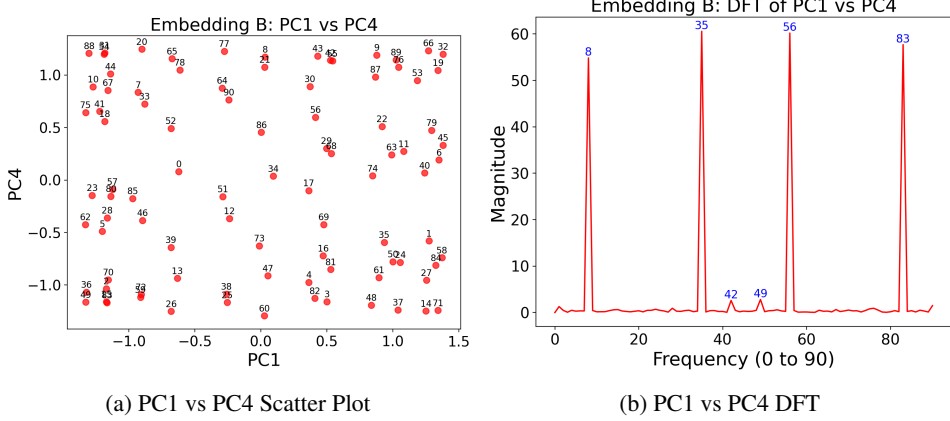

(a) PC1 vs PC4 Scatter Plot

(b) PC1 vs PC4 DFT

Figure 11: PCA and DFT for PC1 ($f = 35$) vs PC4 ($f = 8$), a non-circular embedding.

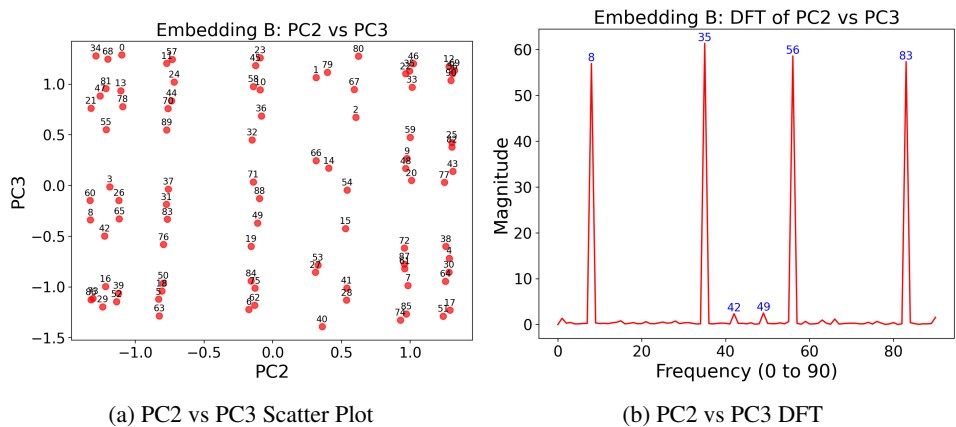

(a) PC2 vs PC3 Scatter Plot

(b) PC2 vs PC3 DFT

Figure 12: PCA and DFT for PC2 ($f = 35$) vs PC3 ($f = 8$), a non-circular embedding.

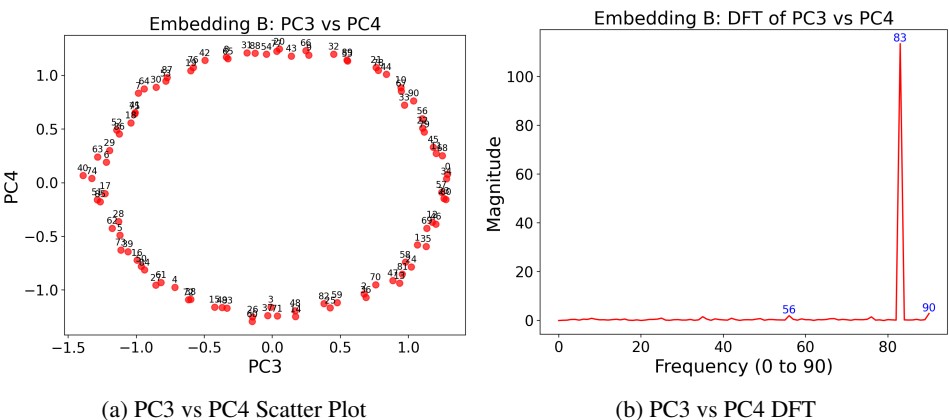

(a) PC3 vs PC4 Scatter Plot

(b) PC3 vs PC4 DFT

Figure 13: PCA and DFT for PC3 ($f = 8$) vs PC4 ($f = 8$), which is a circular embedding because both PC's come from the same frequency cluster.

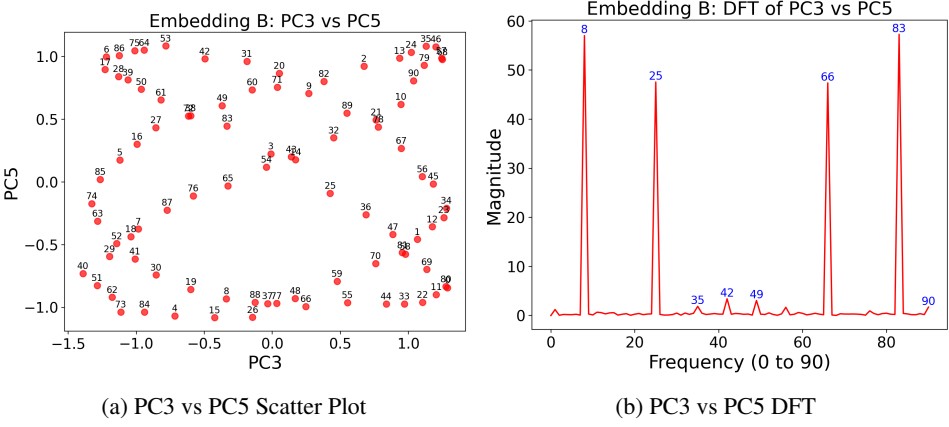

(a) PC3 vs PC5 Scatter Plot

(b) PC3 vs PC5 DFT

Figure 14: PCA and DFT for PC3 ($f = 8$) vs PC5 ($f = 25$), a non-circular embedding.

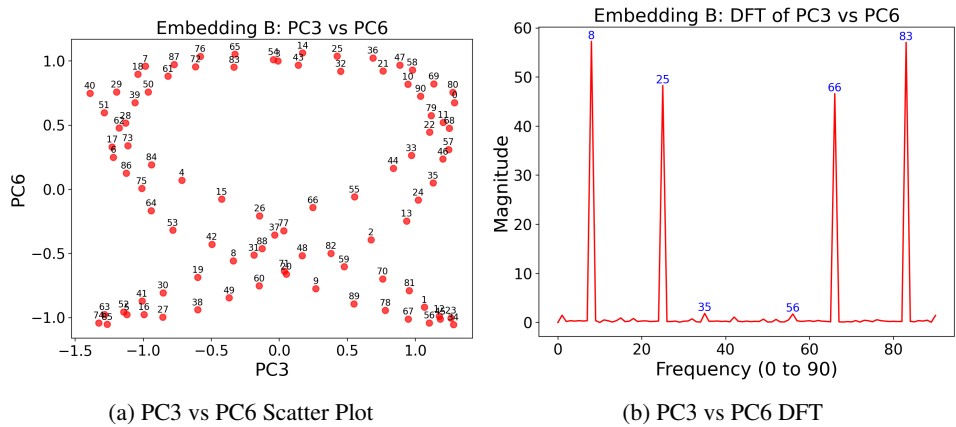

(a) PC3 vs PC6 Scatter Plot

(b) PC3 vs PC6 DFT

Figure 15: PCA and DFT for PC3 ($f = 8$) vs PC6 ($f = 25$), a non-circular embedding

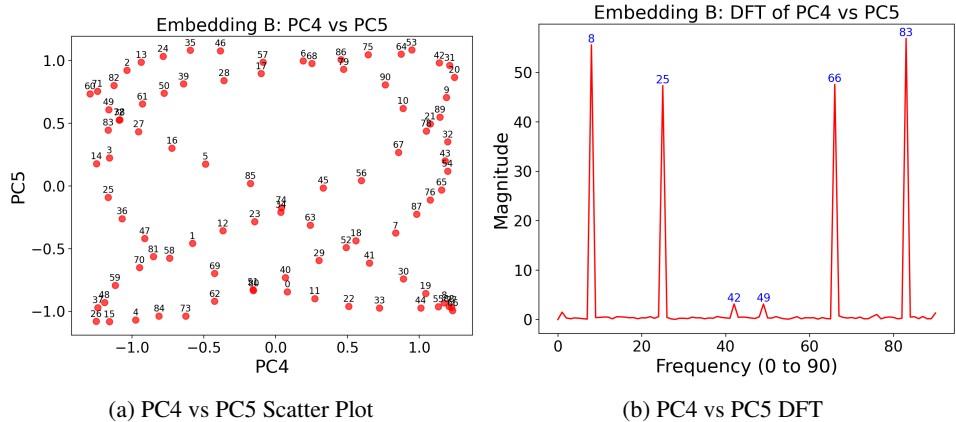

(a) PC4 vs PC5 Scatter Plot

(b) PC4 vs PC5 DFT

Figure 16: PCA and DFT for PC4 ($f = 8$) vs PC5 ($f = 25$), a non-circular embedding.

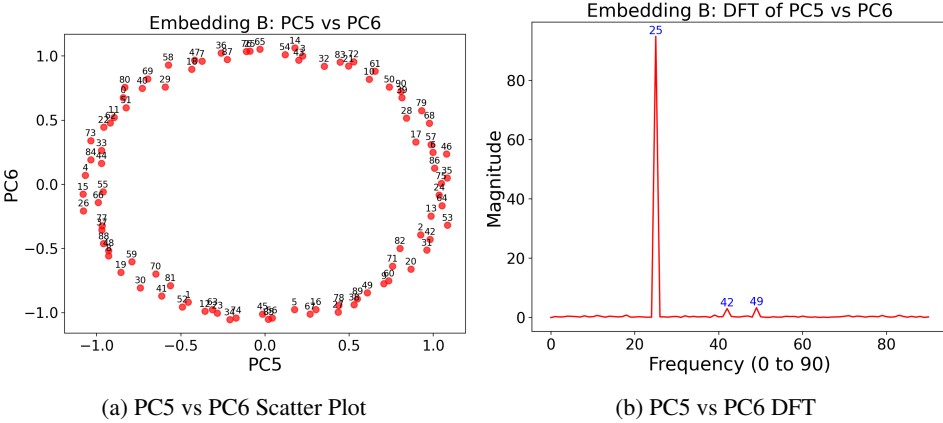

(a) PC5 vs PC6 Scatter Plot

(b) PC5 vs PC6 DFT

Figure 17: PCA and DFT for PC5 ($f = 25$) vs PC6 ($f = 25$), a circular embedding as both PCs come from the same frequency cluster.

### B.1.1 PIZZA CLUSTERS LEARN THE APPROXIMATE CHINESE REMAINDER THEOREM TOO

We take model A, specifically model_p99zdpze5l.pt, from Zhong et al. (2024) and make Figure 18, which shows that pizzas also output on approximate cosets and perform an approximate CRT just like clocks. Note for example, that the output logits for the cluster with max freq = 15: has maximum activation along an approximate coset $\frac{59}{15} = 3.93$, and if the neuron activates strongly at $a$ then it also activates strongly at $a \pm 4$.

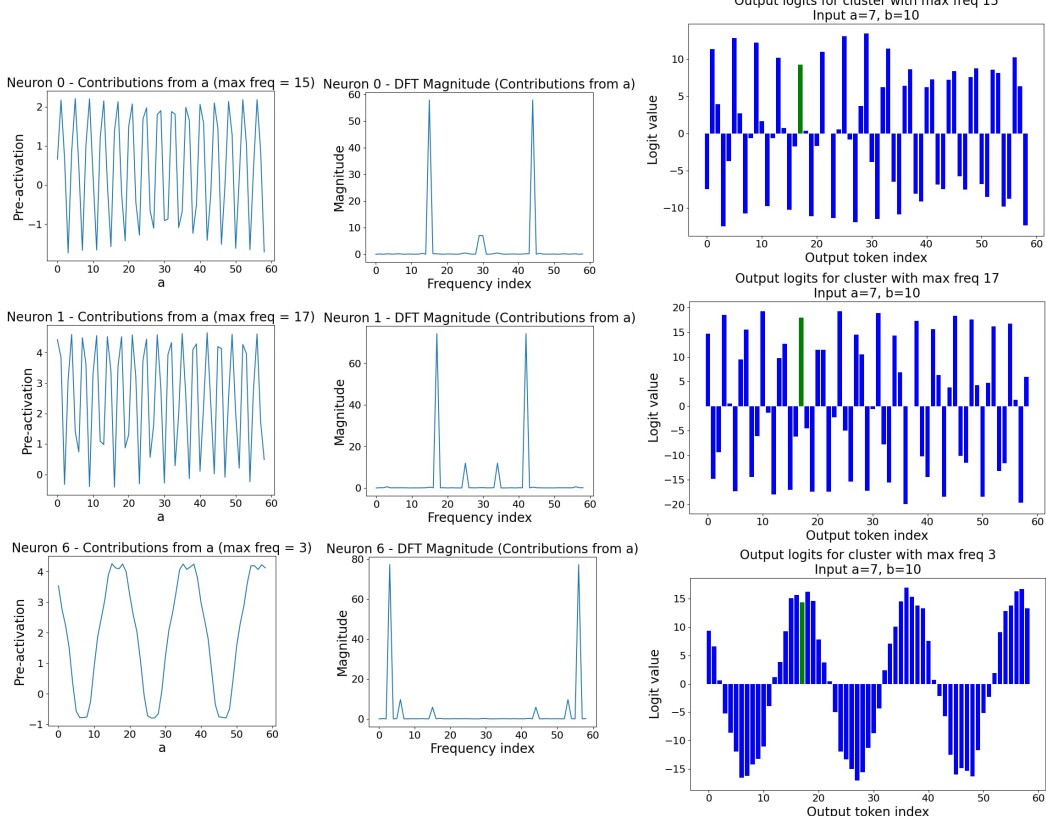

Figure 18: This figure shows three neurons and their DFT's, each from one of three clusters in model A, from Zhong et al. (2024) for these experiments. Note that the pizza neurons (and clusters) are also implementing the abstract approximate Chinese Remainder Theorem algorithm, despite their low level differences with clocks.

Furthermore, consider that remapping the pizza neurons makes their behavior look almost identical to simple neurons when they are remapped, see Figure 19.

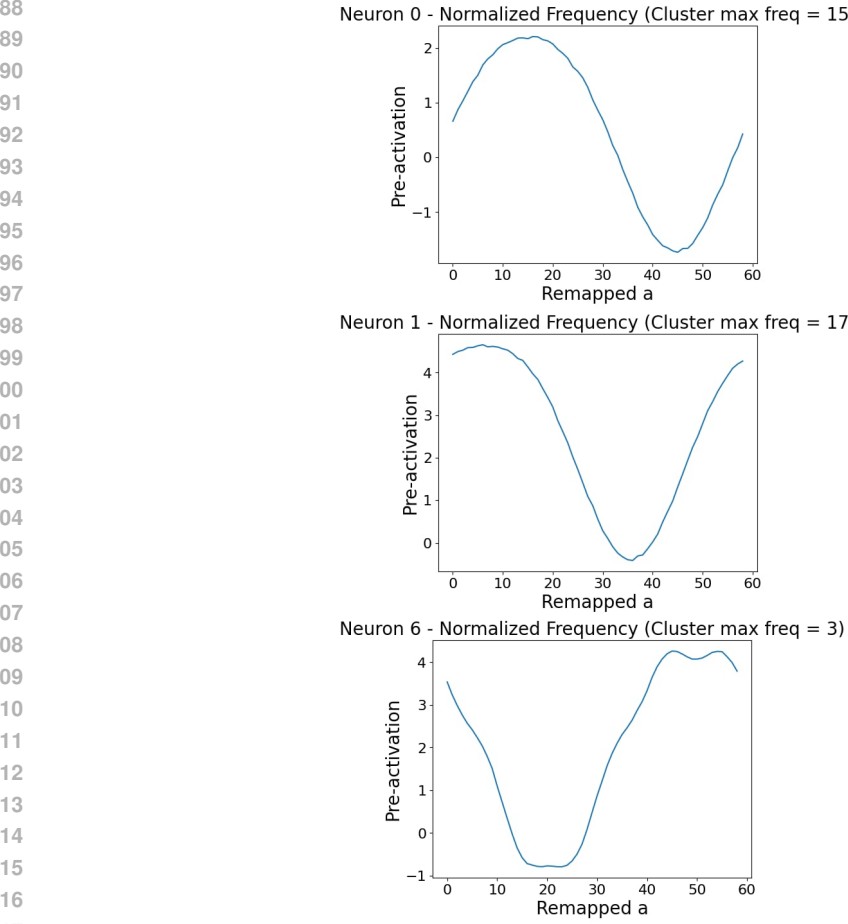

Figure 19: Remapping the pizza neurons shown in Figure 18 shows that they look identical to simple neurons.

### B.1.2 FINE-TUNING NEURONS ARE COMPOSED OF LINEAR COMBINATIONS OF REPRESENTATIONS

We train a neural network with random seed 133 and discover a cluster of fine-tuning neurons. The preactivations for two of these neurons are shown in Fig. 20 and the DFT's for these two neurons are shown in 21. We show that these neurons can be generated by linear combinations of representations in Fig. 22.

**Cluster 3**

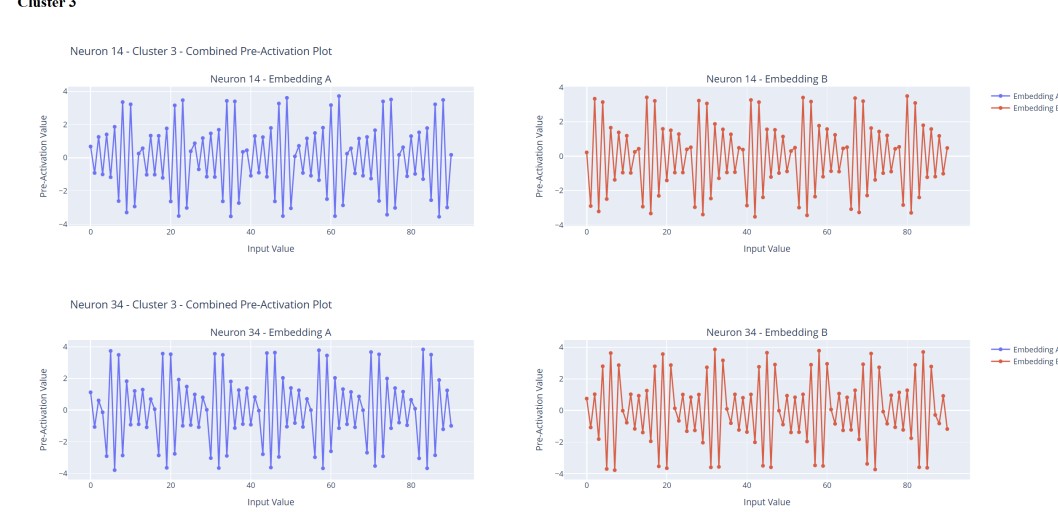

Figure 20: This shows a cluster of fine-tuning neurons and shows the preactivations of the first two neurons in the cluster. The x-axis is the input value into the network for $a$ on the left, and the input value for $b$ on the right.

**Cluster 3**

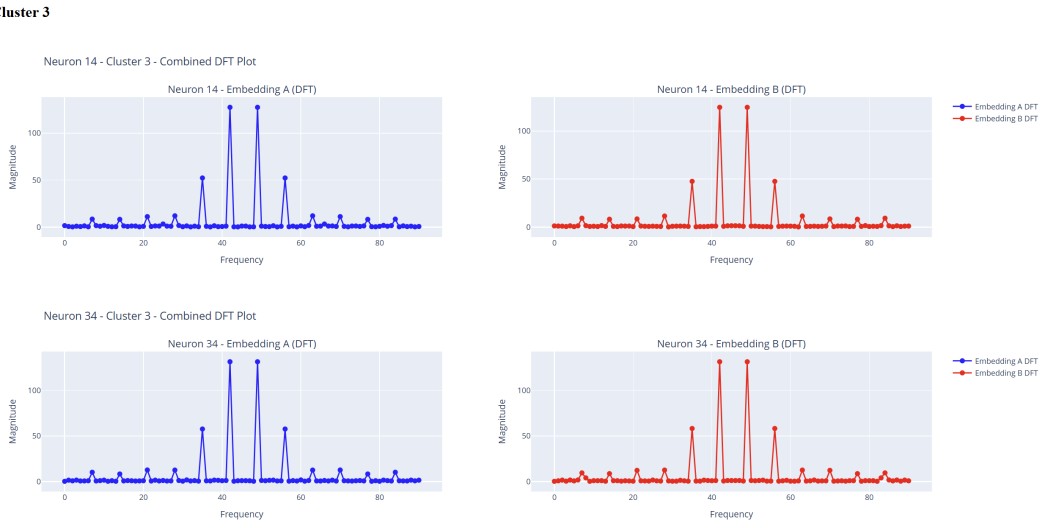

Figure 21: This shows the DFT's of the preactivations of the fine-tuning neurons seen in Fig. 20. The x-axis is the frequency (from 0-90 because this is $(a + b) \mod 91$. The y-axis shows that the representations contributing are $42, 35, 28, 21, 14, 7$ in descending order. Note the DFT is symmetric about its midpoint so values after 45 contain the same information as the values up to 45.

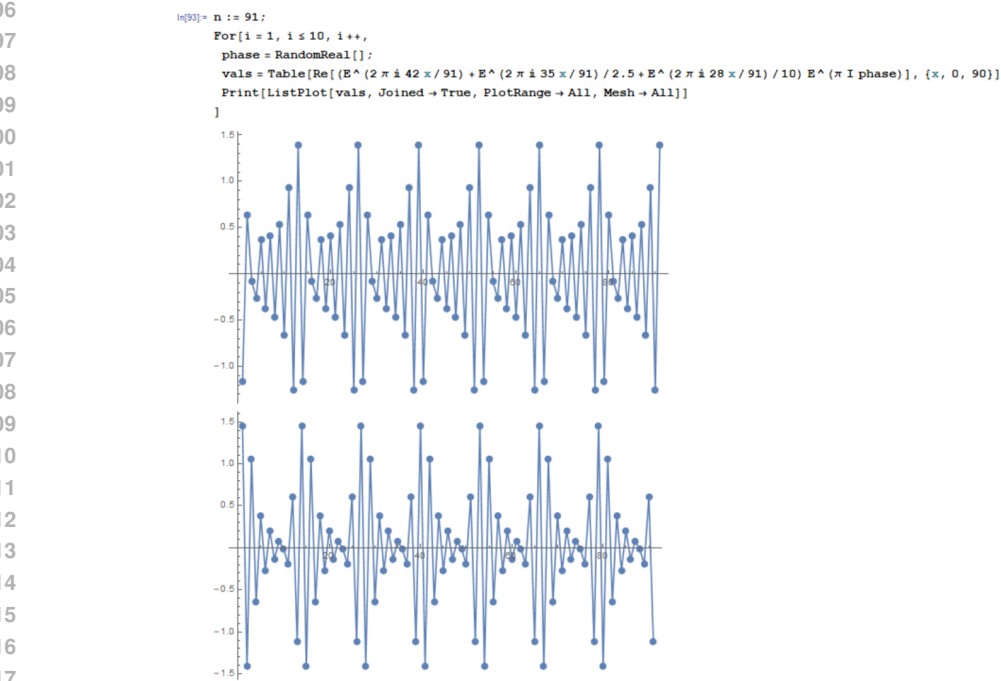

Figure 22: Constructing a fine-tuning neuron. This diagram illustrates the step-by-step process of constructing a fine-tuning neuron, highlighting that it is a linear combination of representations.

## B.2 A CLUSTER OF SIMPLE NEURONS

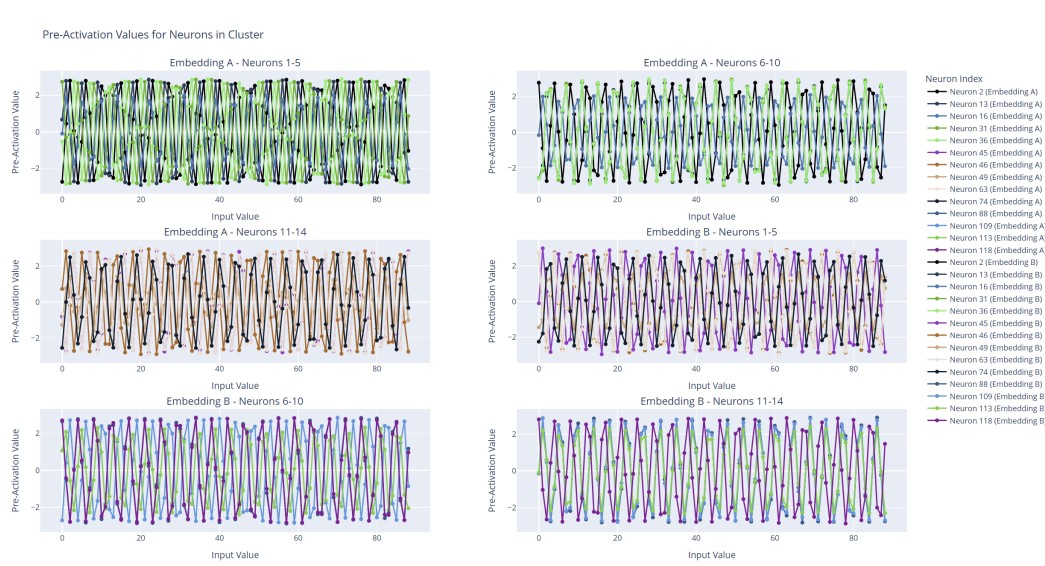

Figure 23: an example cluster of 14 simple neurons of frequency 21.

### B.2.1 REMAPPING EVERY NEURON IN THE CLUSTER TO PERIOD 1 BY APPLYING A GROUP ISOMORPHISM

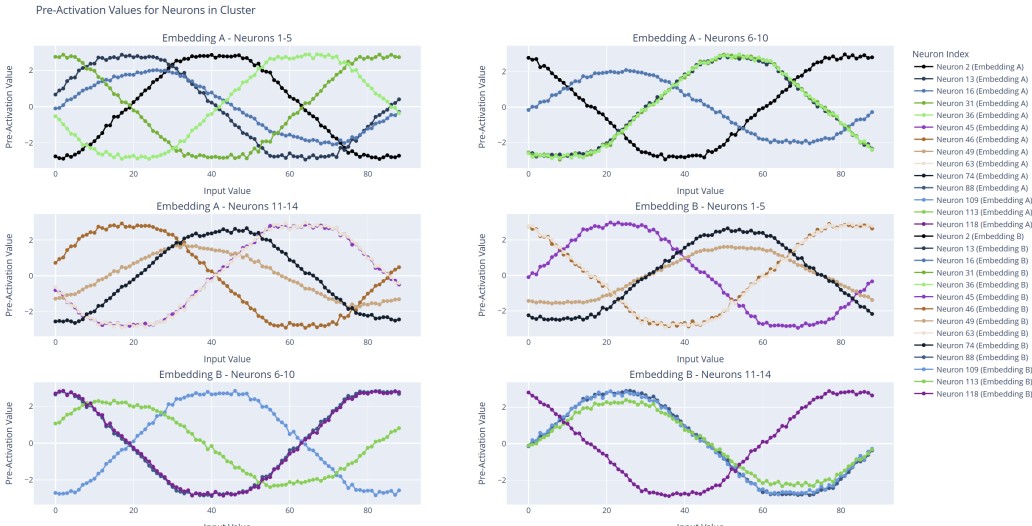

Figure 24: A cluster of simple neurons (from Fig. 23) transformed so that all neurons have period 1.

### B.3 INSPECTING THE PHASE SHIFTS OF THE PERIODIC FUNCTIONS LEARNED BY SIMPLE NEURONS

Here we show how the phases of different neurons in a cluster overlap to give some more information about how clusters of neurons function. See Fig 25a for the histograms of the phases of the preactivations of the neurons in a cluster.

For a higher resolution view of what's going on, see a 2d scatter plot created by grouping the phases for each neuron's $a$ and $b$ preactivations into a pair (phase-a, phase-b) and plotting the points for all neurons in the cluster in the 2d plane as a black point, see Fig 25b. It's worth noting that the phases are nice and spread out uniformly like in Fig 25b only about half the time.

Cluster 1: 14 neurons, frequency = 21

Histogram of Phases for Cluster 1; 14 neurons, frequency = 21

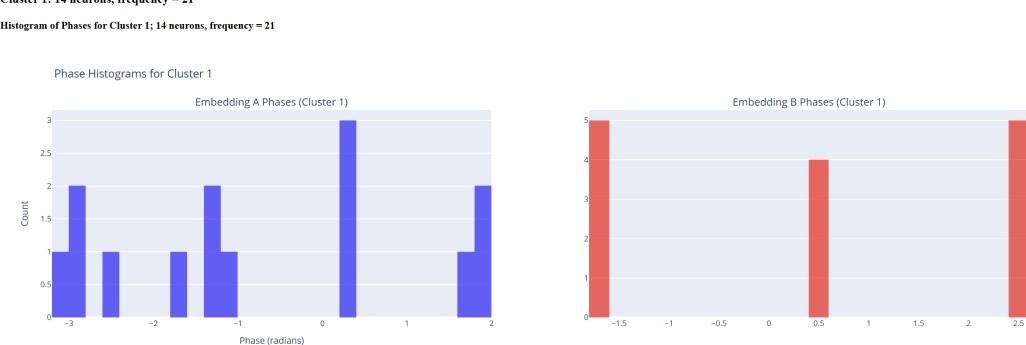

(a) This shows a cluster of fine-tuning neurons and shows the preactivations of the first two neurons in the cluster. The x-axis is the input value into the network for $a$ on the left, and the input value for $b$ on the right.

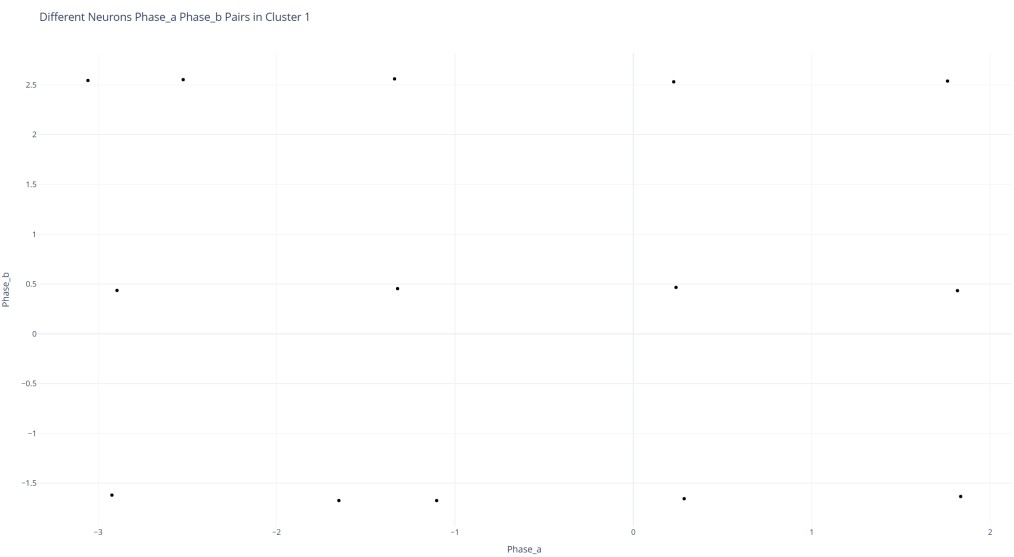

(b) This shows a 2d scatter plot created by grouping the phases for each neurons $a$ and $b$ preactivations into a pair (phase-a, phase-b) and plotting the points for all neurons in the cluster in the 2d plane as a black point. In this case, the cluster has 14 neurons of frequency 21.

Figure 25: Figures showing the histogram of phases, then the 2d scatter plot of phases for all 14 neurons in a simple neuron cluster of frequency 21.

### B.3.1 HISTOGRAMS RECORDING AN UPPER BOUND ON THE COUNTS OF SEEING FINE-TUNING CLUSTERS WITH A GIVEN FREQUENCY

Note that the next two histograms are created by recording frequencies with weights in the DFT in the range of (7.5, 30). This is not a sufficient way to always detect fine tuning neurons, and sometimes it will include simple neurons in its counts, however this is much more rare. If you consider the ability for neurons with preactivations of specific frequencies to contaminate other neurons frequencies slightly (because they may modify values in the embedding matrix by a small amount), you will see where this counting method can go awry. It is however the case that usually, the contamination coming from a different cluster of simple neurons is below 7.5. Thus, these plots should not be considered "accurate" and just approximations.

These plots are still useful to show the relative frequency of simple neurons vs. fine tuning neurons. The histogram of Frequencies found Fig. 4a found a uniform distribution with each frequency showing up about 10k times. Removing the vast majority of contamination by filtering with 7.5 (usually the DFT magnitudes on other frequencies are 0 and if they aren't near 0 then they are

less than 4 and there is a simple neuron making use of that frequency in a different cluster (*i.e.* a simple neuron has one big spike with magnitude over 60 on that frequency). This gives us about 2200 fine-tuning neurons found with each frequency, including overcounting because fine-tuning neurons make use of linear combinations of representations and thus their DFT usually has three or more values in the range (7.5, 30). Thus the histograms of frequencies associated with fine-tuning neurons are upper bounds on the number of clusters that are identified across 100k random seeds to be fine-tuning neurons. Assuming the upper bound is reality and no overcounting occurs (which it does), we would get about $43 \times 2200 = 94600$ clusters of fine tuning neurons found in 100k training runs. Comparing this to the cluster frequencies histogram Fig. 4a, which shows about $43 \times 11500 = 494,500$ total clusters learned in 100k training runs. This includes fine tuning neurons, so the empirical probability of observing a cluster of fine tuning neurons is at most around: $\frac{94600}{494,500-94600} =\sim 0.23656$. In reality it is smaller due to overcounting.

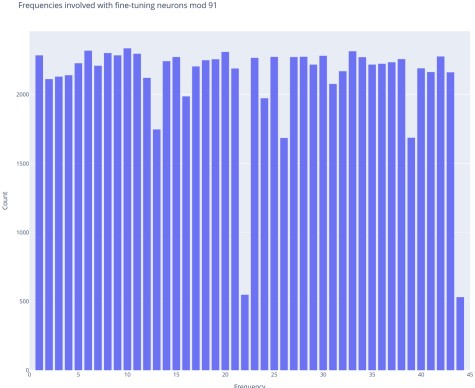

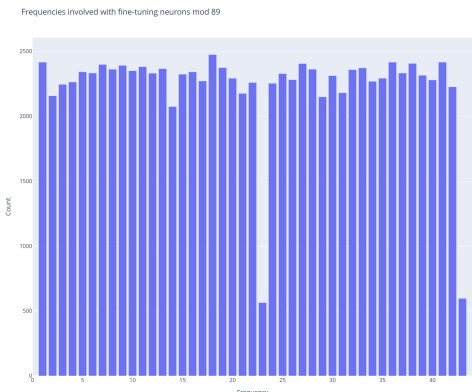

(a) The histogram of frequencies associated with fine-tuning neurons over 100k random seeds mod 91. Note that 22 and 44 are least likely, with 13, 26, 39 (prime factors) being less likely.

(b) The histogram of frequencies associated with fine-tuning neurons over 100k random seeds mod 89. Note that 23 and 43 are least likely

Figure 26: Comparison of histograms of frequencies associated with fine-tuning neurons over 100k random seeds for mod 91 and mod 89.

## B.4 NOISE AND ABLATION PLOTS

In this section we take the clusters from random seed 133 and we randomly inject multiplicative scaling noise into every weight attached to neurons in the cluster. We do this by multiplying the weight by $e^s$, $s \sim \mathcal{N}(0, \sigma)$, for $\sigma$ in [0.]

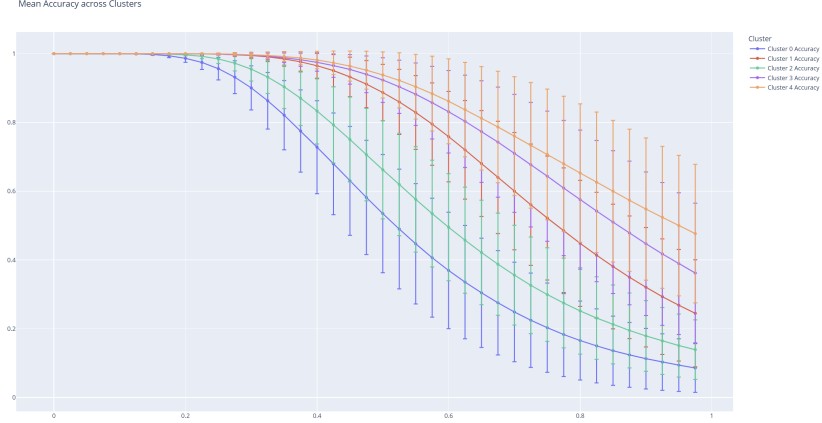

(a) Multiplicative Noise injected into every weight of every neuron in a cluster from a normal distribution with std dev $\sigma$.

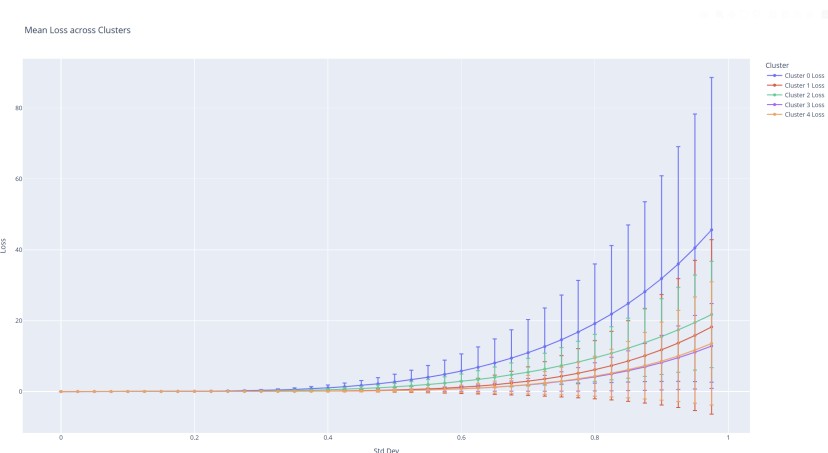

(b) Multiplicative Noise injected into every weight of every neuron in a cluster from a normal distribution with std dev $\sigma$.

Figure 27: Note the neural network is robust to quite large amount of noise being injected. The loss barely changes with when using a std dev of 0.225, which is a strong multiplicative scaling factor.

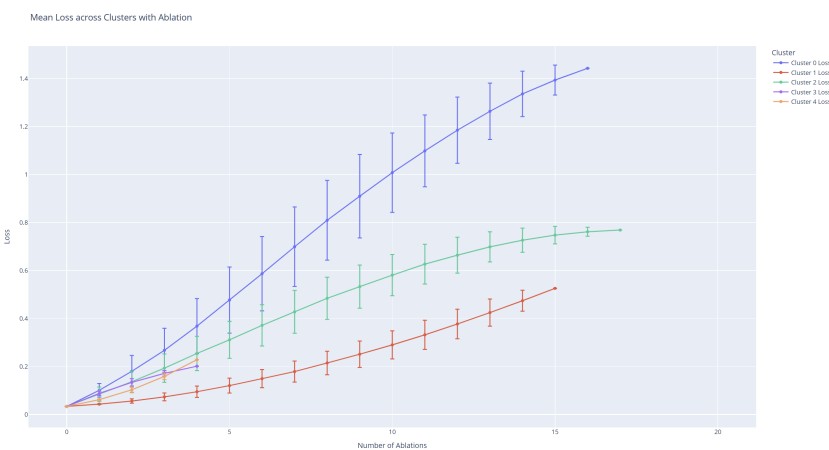

(a) Ablation study showing the impact on the loss function with the removal of random neurons in specific clusters of the network.

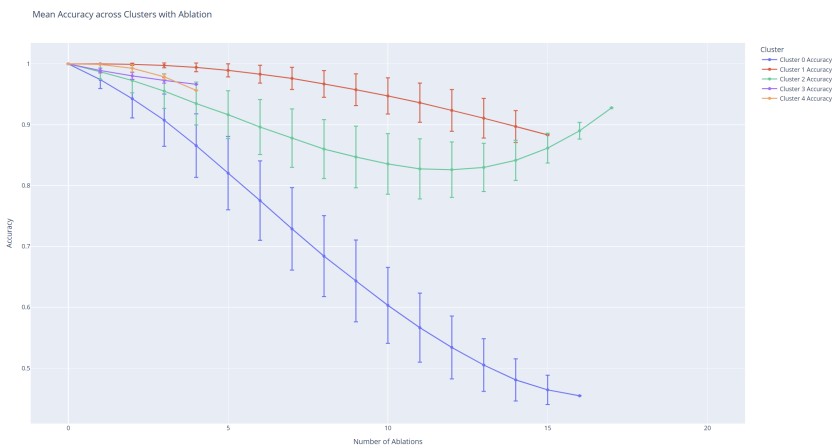

(b) Ablation study showing the impact on accuracy with the removal of random neurons in specific clusters.

Figure 28: Results of the ablation study. The loss and accuracy metrics highlight the influence of removing neurons randomly from a cluster, showcasing the importance of neurons within each cluster

## B.5   FINE-TUNING NEURONS LIKE ADDITIVE AND SUBTRACTIVE RELATIONS

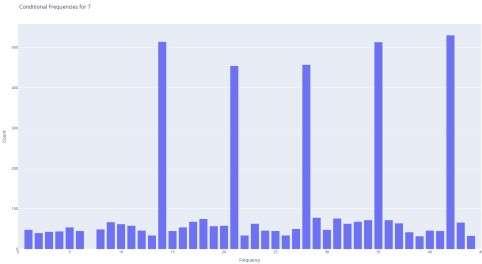

(a) Fine-tuning Neuron Additive Relations given 15 is a frequency

(b) Fine-tuning Neuron Additive Relations given 7 is a frequency

Figure 29: Side-by-side figures showing the fine-tuning neuron additive relations for two different cases: if a neuron with frequency 15 is learned, frequencies that are multiples of 5 are all more likely to be found. This is also true for 7, which is a prime factor of 91, which is the moduli.

## B.6 SCALING THE NUMBER OF NEURONS IN THE LAYER ACHIEVES EXPERIMENTAL RESULTS WITHIN $\mathcal{O}(\log(n))$

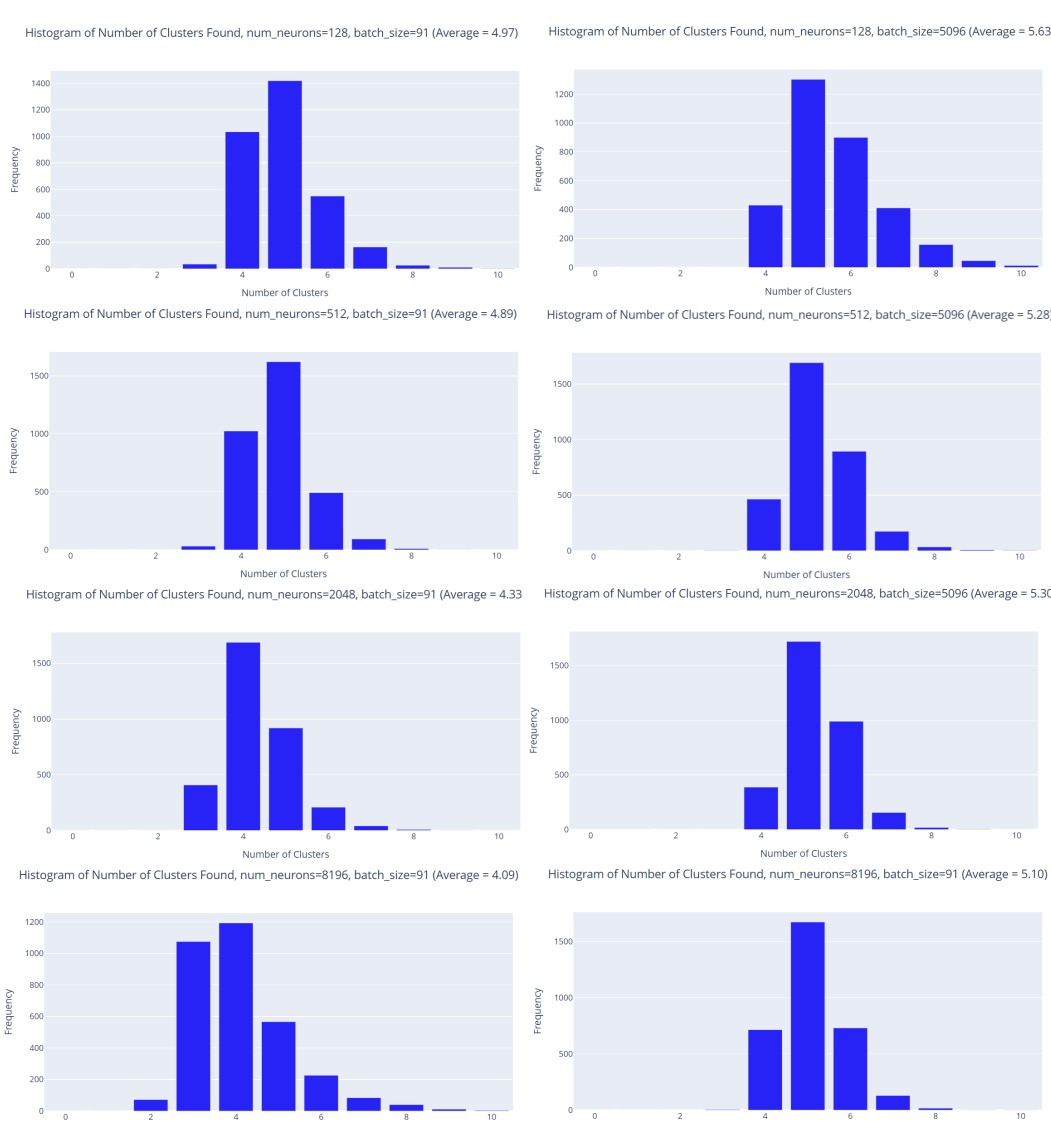

Figure 30: This figure shows that the scaling is always $\mathcal{O}(\log(n))$, even as the number of neurons is increased from 128, to 512, to 2048, to 8196. The first column is batch_size=91 and the second column is batch_size=5096, *i.e.* the entire training set size. All results are $\mathcal{O}(\log(n))$.

## B.7 SCALING THE MODULI OF THE DATASET ACHIEVES EXPERIMENTAL RESULTS WITHIN $\mathcal{O}(\log(n))$

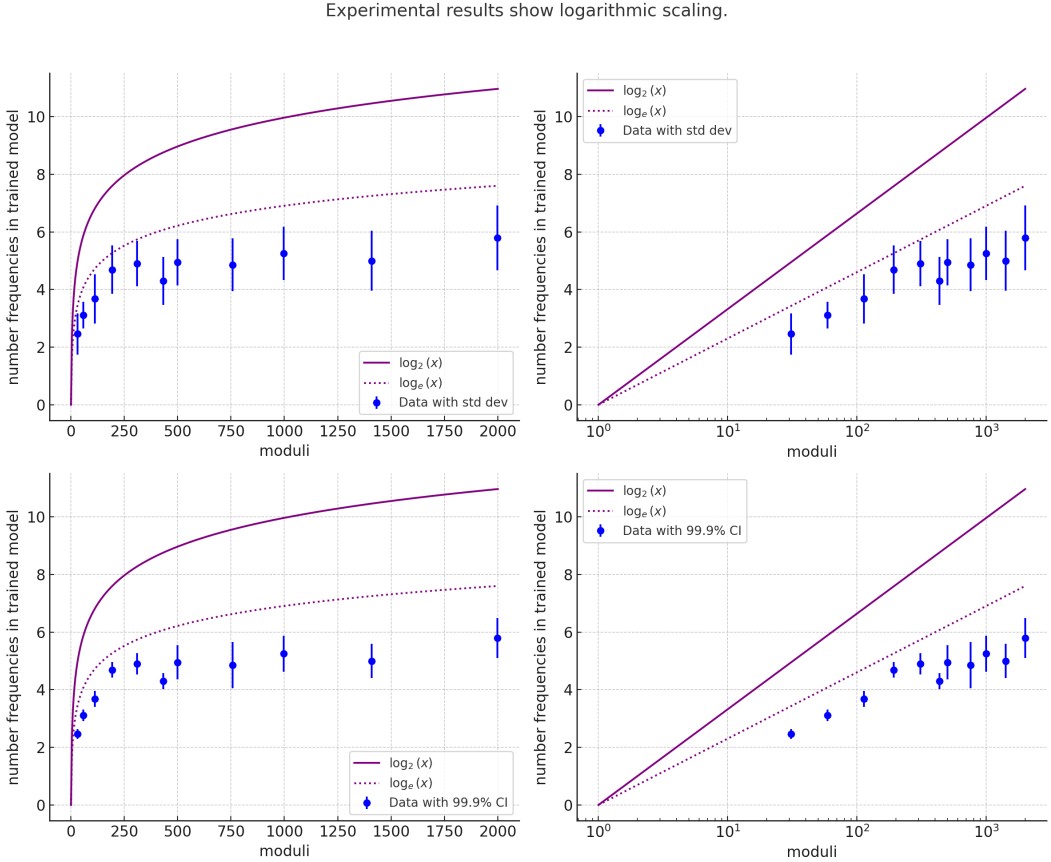

Figure 31: This figure shows that the experimental scaling is reasonably upper bounded by $\log(n)$ for networks trained on $(a + b) \mod n$, agreeing with the $\mathcal{O}(log(n))$ conjecture. This plot was made by training 200 neural networks at each of the points, and taking the average and std deviation of the number of clusters.

| Number | Learning Rate | Batch Size | Weight Decay | Training Set Size |
|--------|---------------|------------|--------------|-------------------|
| 59 | 0.008 | 59 | 0.001 | 1770 |
| 113 | 0.004 | 113 | 0.0003 | 6780 |
| 193 | 0.003 | 193 | 0.0001 | 18914 |
| 310 | 0.0008 | 310 | 0.00008 | 46500 |
| 433 | 0.0006 | 433 | 0.00005 | 86600 |
| 499 | 0.0005 | 499 | 0.00003 | 124750 |
| 757 | 0.0003 | 757 | 0.0000085 | 280090 |
| 997 | 0.0003 | 997 | 0.0000015 | 498500 |
| 1409 | 0.00028 | 1409 | 0.0000009 | 986300 |
| 1999 | 0.00024 | 1999 | 0.0000008 | 2398800 |

Table 2: Experimental results with Adam optimizer across varying parameters for Figure 31.

## C  SHOWING THAT IF YOU SHIFT (A,B) BOTH BY 2, THE CLUSTERS SHIFT BY 4

Here you can see that the clusters of neurons are approximately equivariant to shifts in the inputs, *i.e.* the cosets shift with the inputs.

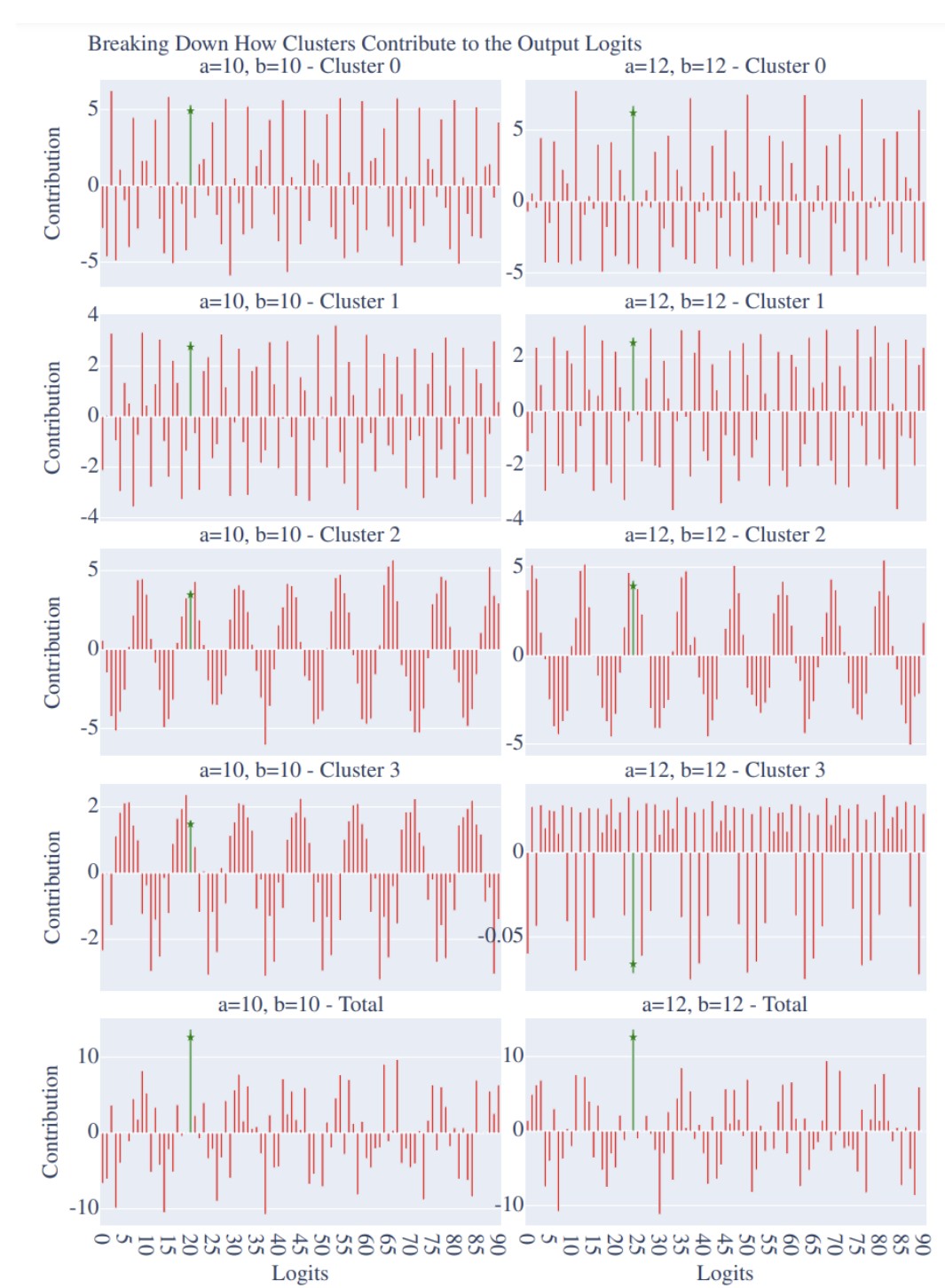

Figure 32: Here you can see that the clusters of neurons are approximately equivariant to shifts in the inputs, *i.e.* the coset clusters shift with the inputs, implying the network has learned cosets that it's utilizing to intersect, via linear combinations, to perform the approximate CRT. This example was chosen to show that the network did not learn a global minimum, *i.e.* cluster 3 has only 4 neurons, which does not grant it enough expressivity to always be equivariant. Cluster 0 has frequency 35, making it a coset, Cluster 1 has frequency 25, making it an approximate coset, Cluster 2 has frequency 8, making it an approximate coset, Cluster 3 has frequency 42, making it a coset. This is the same random seed as the ablation study (Fig. 28a); Cluster 0 is doing the most.

