# OpenReview forum: "Interpreting neural networks depends on the level of abstraction: Revisiting modular addition"
_ICLR.cc/2025/Conference — Submitted to ICLR 2025_

### Official Review · Reviewer_8ZKm · 2024-10-26

**Soundness:** 4
**Presentation:** 3
**Contribution:** 2
**Rating:** 6
**Confidence:** 3

**Summary:**

This paper revisits the modular addition and aims to find a unifying answer in light of the divergent narratives in the literature. Their main discoveries are: (1) they study not only prime but also composite p for modulo addition. By scanning a large number of random seeds, they discovered the learned frequencies are uniform (which is expected), but the interaction between frequencies is non-trivial (e.g., if frequency n is learned, then frequency n/2 and 2*n are less likely to be learned). (2) There are simple neurons (pure frequency) and fine-tuning neurons (a mixture of a few frequencies). (3) They proposed a mathematical model which posits that certain neurons represent approximate cosets, unifying clock/pizza algorithms. (4) They derive that O(log n) clusters should be needed, matching with experimental results.

**Strengths:**

1. The paper is well-written and sound, nicely balancing theoretical and empirical results
2. Unifying previous diverging narratives is one contribution of this work
3. They have several very interesting observations that may be worth separate papers: frequency interaction, the distribution of the number of clusters, etc.

**Weaknesses:**

1. Some parts are hard to read. For example, I find myself a bit lost in Section 5.4. It'd be nice to highlight some important takeaways since not all readers can follow through the detailed mathematical derivations.
2. While this paper digs deep into the modular addition task, it is unclear whether/how the discoveries can be extended to general tasks.

**Questions:**

1. I'm not sure what the "level of abstraction" in the title refers to.
2. The result that O(log n) clusters are needed is nice, but since only n = 89 and 91 are studied, it could be a fluke. Is there evidence for smaller n or larger n to have fewer/more number of clusters?

---

> ### Author Response · Authors · 2024-11-21
> **Official Comment by Authors**
>
> Thank you for your thoughtful review and for highlighting the strengths of our work, particularly our ability to unify divergent narratives and provide novel insights into modular addition. We are encouraged that you found our paper well-written and sound.
>
> **Weaknesses**
> 1. Section 5.4 is indeed technical. To aid clarity, we can add a remark that it's similar in spirit to a Probably Approximately Correct bound, which is common in learning theory. The point of the section, however, is that we model what networks were observed to have been doing quantitatively and qualitatively across all of the literature, as being different implementations of approximate cosets and cosets in section 5.2. Then in section 5.4, we put these together mathematically. the same way the networks have been observed to do, and derive an approximate Chinese Remainder Theorem (CRT) algorithm. Doing so we manage to get a mathematical bound of $\mathcal{O}(\log(n))$, which matches all of these results, and was built from the ground up by observing that neurons were learning cosets and approximate cosets. It's possibly worth noting that the CRT is the actual theorem for breaking down modular arithmetic, and it uses $\mathcal{O}(\log(n))$ prime factors to do so, making it likely that we've achieved an upper bound that matches reality.
> 2. Our global rebuttal addresses what general insights come from our paper, most important being that the universality hypothesis seemed false on modular arithmetic before our work, and now it seems likely to be true. *The reason that this is so important is because resolving conflicts in simpler settings like modular addition is a necessary step toward understanding the potential universality of neural networks across all tasks, and modular addition has shown that proposed universal algorithms are beyond microscopic details and tied to interpretations at the correct level of abstraction.* Our work does not provide explicit methods for more general tasks, but it provides evidence on how levels of abstraction matter for interpretations (i.e. the scale at which you inspect the system). This seems even more important for general tasks and larger networks going forward, because if contradictions occur in the setting of modular arithmetic, they will certainly occur in more complex settings. At a minimum, all experimental interpretability work on modular addition falls under our umbrella of them being different implementations of this CRT-like algorithm, using cosets and approximate cosets. We have the added benefit that Stander et al.'s work [1] showed cosets on the permutation group, which means that universality may even be true across totally different datasets (the permutation group is non-abelian and is the most structured finite group, whereas the cyclic group is abelian and has the least structure). Thus, the point is that every paper that has followed Nanda et al.'s [2] seminal paper on this subject is unified under our cosets interpretation.
>
> It's worth mentioning that the reason the universality hypothesis is so exciting is because of its potential applications should it end up being true. If neural networks trained on different data still learn structures that are similar at an abstract level, it will greatly aid work on the safety of AI systems and automated circuit detection methods. Furthermore, we rewrote a paragraph in our discussion and conclusion, which poses the question: ``should we consider an interpretation to be good only if it's true at more than one scale of abstraction?''. The reason that the universality hypothesis appeared to be convincingly false was because all the literature since Nanda et. al. [2] found different microscopic algorithms, yet we were able to tie everything together, at three different scales of abstraction, under the view of each preceding paper being different implementations of cosets and approximate cosets, that come together to form an overall approximate CRT-like algorithm.

---

> > ### Author Response · Authors · 2024-11-21
> > **Official Comment by Authors**
> >
> > **Questions**
> > 1. We added a figure and rewrote the abstract to clarify what we meant by levels of abstraction. It's now on page 1 and shows that the current literature bases their interpretations on microscopic individual neuron behaviour, or principal components. Nanda et al. did mention constructive and deconstructive interference, but focused on presenting an algorithm explaining why the correct logit was maximized, and not why the incorrect logits were minimized, leaving the abstract picture to the future. Gromov [3] then presented a brute force algorithm explaining the constructive and deconstructive interference by using $\mathcal{O}(n)$ frequencies. Our interpretation can be interpreted as cosets being learned at the individual neuron level, the cluster level, and at the total network level where $\mathcal{O}(\log(n))$ cosets are intersected via linear combinations of positive and negative contributions in order to minimize the loss by maximizing the correct logit, while simultaneously minimizing incorrect logits.
> > 2. The $\mathcal{O}(\log(n))$ is likely not a fluke because it tightly bounds all pre-existing experimental results in the literature. We fix $\rho$ and $\delta$ and do not change the values for our experiments, and the bound continues to match the experimental results. For the time being however, our $\mathcal{O}(\log(n))$ bound is substantially better than the next best model in the literature, which uses $\mathcal{O}(n)$ frequencies (Gromov). Furthermore, we ran scaling experiments, please see Fig. 29 and Fig. 30 in the appendix of our revision, which both add a lot of data points supporting this bound.
> >
> > [1] Stander et al., "Grokking Group Multiplication with Cosets" (2023)
> >
> > [2] Nanda et al., "Progress measures for grokking via mechanistic interpretability" (2023)
> >
> > [3] Gromov, "Grokking Modular Arithmetic" (2023)

---

> > > ### Comment · Reviewer_8ZKm · 2024-11-24
> > >
> > > I would like to thank the authors for their thoughtful responses. I find the new scaling experiments in Figure 30 interesting (the error bars obscure the trend a bit, but, understandably, initializations and batch noises can impose large uncertainty in the number of frequencies). Overall, I remain optimistic about this paper. While I agree with other reviewers that this paper may not be completely novel, their experimental results are solid and interesting. I'm willing to keep a positive score for their hard work on comprehensive experiments and a number of interesting observations (e.g., f and f/2 interference, log(n) clusters etc.), but I agree with other reviewers that interpretations of these observations need to be more careful (either tone down the claims, or add more evidence to support the claims).

---

> > > > ### Author Response · Authors · 2024-11-25
> > > > **Official Comment by Authors**
> > > >
> > > > Thank you for your positive response! We ask you to please consider the following points about why our work is novel and significantly different to previous works; we also further explain the log(n) result below:
> > > >
> > > > 1. All of the preceding work (Nanda et al., Chughtai et al., Gromov, Zhong et al., Stander et al.) are in contradiction--they find different interpretations and propose models that fail to be universal across architectures and hyperparameter settings. Our paper unifies these works, showing that they actually all observed different implementations of a universal algorithm no one suspected was being learned. We believe this is a large contribution due to generalizing every paper and all their results on this topic under one unifying theory. We added Fig. 17, taking the saved pizza model from Zhong et al., to make it clear that pizza networks also learn approximate cosets and an approximate Chinese Remainder Theorem (CRT) (it is the same CRT algorithm as Fig. 6 and 31 which are clocks). Across all networks in the literature and throughout the 1 million models that we trained, across different hyperparameters and even architectures, all learned cosets or approximate cosets, and used them in the same CRT-like way as shown in Figures 6, 17 and 31.
> > > > 2. Elaborating on CRT: No one has ever suggested cosets or approximate cosets for modular arithmetic because it is not obvious (or expected) that neural networks would learn an approximate CRT when trained on modular arithmetic with prime moduli. This is because there are no cosets when training with prime numbers. The neural network instead learns approximate cosets, and uses them like they're cosets to form an approximate CRT. We found this behavior to be remarkable due to our biases from our math knowledge making the CRT seem unlikely for prime moduli.
> > > > 3. Gromov's paper uses (n-1)/2 deterministic frequencies to make the output from the neural network a Dirac delta function on the correct logit. This model does not match any experimental results in the literature, or what we found. All experimental results are sublinear, and on the order of $O(log(n))$, within a (small) constant factor.
> > > > 4. Experimentally: as the width is increased, the number of clusters found by the network decreases (see Fig. 29 in our appendix). Our model predicts this behavior as the neural network finds lower loss solutions, with O(log(n)) tightness. Gromov's theoretical model uses all frequencies to construct a Dirac output on the logits, and as width is increased, retains its usage of all frequencies, but this isn't observed empirically.
> > > > 5. The $O(log(n))$ bound is a falsifiable theory that, at the time of our presentation, aligns with all known experimental results. Our model presents that if (and this is the assumption) the neural network is seeking to obtain a certain size of separation (difference between largest output logit and second largest output logit), with reasonably high confidence, then the network must use between $\Omega(log(n))$ and $O(log(n))$ number of frequencies. To clarify: we show that a "separation of at least delta*m occurs with probability at least rho" is equivalent to "m > [expression]". So what does this mean? First it means that if $m$ is much less than this expression, we are unlikely to get the desired separation. This is the lower bound--we need at least $\log(n)$ to get the desired separation. Secondly, it also means that if $m$ is even just a bit bigger than the expression (e.g. twice as big - in particular still O(logn)) then we do get the desired separation. Thus we can claim that the number of frequencies it needs to learn is bounded above and below by a constant times $\log(n)$. This doesn't mean that the neural network will not learn more frequencies! Indeed, this is where all experimental results in the literature, plus our own come in, and show that the number of frequencies is always in fact observed to be $O(\log(n))$. In the event that the network wants to further minimize the loss, it could eventually converge to Gromov's solution, which again, networks trained with stochastic gradient methods have never been observed to learn. Please see https://postimg.cc/d78mQbP1 for a comparison of how Gromov's model is scaling compared to our experimental results. Note how tiny the error bars become when there is a linear function on the plot due to their relative size (you will have to zoom in). We are running new experiments because we suspect the std deviation grew because the moduli (>=499) became the same size (or even larger) than the network width, which was 512. Even if the new results still have the large std deviation, we believe we have presented reasonable evidence that the growth rate is sublinear--a substantial improvement on current literature.
> > > >
> > > > We hope this addresses your remaining concerns; if so, we would be grateful if you would consider increasing your score. Otherwise, we are eager to discuss further! Thanks again for your time.

---

> > > > > ### Author Response · Authors · 2024-12-02
> > > > > **Response by the authors**
> > > > >
> > > > > Thanks again for your comments! Please see the new scaling plot: https://postimg.cc/0K973RqB. It scales out to 4999 and we check linear, square root, and logarithmic fits, getting $R^2$ values of:
> > > > >
> > > > > | Fit      | $R^2$      |
> > > > > |----------|----------|
> > > > > | Logarithmic  | 0.971  |
> > > > > | Square root  | 0.741  |
> > > > > | Linear  | 0.473  |
> > > > >
> > > > > We also submitted a revised paper that addresses doubts about the novelty of our work (that was questioned by other reviewers). The papers of Gromov and Morwani et al., both study networks with no biases. This is why their theoretical models predict $\frac{n-1}{2}$ and experimental results find $\frac{n-1}{2}$ frequencies, whereas our model is for the more general case of when networks are trained with biases, and it predicts $\mathcal{O}(log(n))$ frequencies, as backed by our scaling plot out to 4999.

---

### Official Review · Reviewer_qgqV · 2024-11-01

**Soundness:** 2
**Presentation:** 2
**Contribution:** 2
**Rating:** 5
**Confidence:** 3

**Summary:**

The paper focuses on how neural networks learn modular addition with prime and composite modules. The authors show that different previously observed algorithms are unified manifestations of the same underlying mechanism, which is related to implementing coset structures of subgroups under addition. They identify two types of neurons (simple and fine-tuning) and demonstrate that networks implement an approximate version of the Chinese Remainder Theorem by clustering neurons with different frequencies.

**Strengths:**

1. Clear and straightforward setup
2. The authors offered simple mathematical explanations of the phenomenon they observed
3. The authors tried to build a unified view on different algorithms observed related to the ``grokking'' phenomena

**Weaknesses:**

1. Several references were overlooked or inadequately credited, which affects this paper's novelty:

  - D. Stander et al. (arXiv:2312.06581) demonstrated that their model learns coset structures during training on permutation groups $S_5$ and $S_6$, where they also identified circuits using Group Fourier Transform. Their work performed a very similar analysis to this paper's examination of modular addition.
  - The weight structure discussed in Section 5.1 is identical to that proposed by A. Gromov (arXiv:2301.02679). The authors neither properly credit this previous work nor rigorously explain how such weight matrices would correctly generate predictions with ReLU activation. Section 5.2 merely presents minor generalizations to the case of composite numbers.

2. The authors draw conclusions based solely on their specific experimental settings. While they attempt to explain and unify various phenomena, these were originally observed in different settings. The paper would be more convincing if it provided experimental evidence from the original settings used by others and demonstrated that the coset explanation holds consistently across different settings for the same task.

**Questions:**

Could the authors address my concerns mentioned in weakness section?

---

> ### Author Response · Authors · 2024-11-21
> **Official Comment by Authors**
>
> Thank you for your review!
>
> **Weaknesses:**
>
> 1. **a)** Yes we perform a similar analysis to Stander et al. [1], but we come to wildly different conclusions as we study a different problem that, until now, was thought to have provided evidence in the literature against the universality hypothesis. Stander et al. did not reconsider modular arithmetic, and thus could not have re-opened the universality hypothesis. In fact, at the time of its publication, Stander et al. provided additional evidence against the universality hypothesis due to totally different algorithms being found on two different finite groups (cosets on permutation group and GCR on modular arithmetic). Furthermore, the theme of their paper is that "we need to be careful about making claims that proposed interpretations are correct" and we build upon this theme by showing that all of the interpretations for modular arithmetic are correct at the microscopic level, but they are not in contradiction to each other, as each microscopic analysis gave a different implementation of what the network was actually doing -- the approximate Chinese Remainder Theorem (CRT) on approximate cosets. Crucially, studying clusters of neurons, which was not required or suggested by the symmetric group analysis in Stander et al.'s paper, allowed us to identify the (approximate) coset structure for modular addition. **b)** We will add a citation to Gromov [3] in section 5.1 where we give our definition of a simple neuron. Furthermore, Gromov's work provides a $\mathcal{O}(n)$ bound on the number of frequencies required, which is substantially above all experimental results. Our work provides a $\mathcal{O}(\log(n))$ bound, which fits all previous experimental results in the literature and fits all of our experimental results. Our theoretical work is therefore totally different to Gromov's (apart from the first definition used in the proof being the same, as you point out), as we achieve a bound that is an exponential order of magnitude tighter. Furthermore, section 5.2 does not present a minor generalization to only the case of composite numbers. It presents major generalizations to both the prime and composite cases. It was in fact this realization -- that the overall algorithm the network is implementing is the exact same, whether the network learns frequencies that are prime factors of a composite number, or not prime factors -- that gives us the overall generalization that generalizes all preceding literature being the approximate CRT. Section 5.2 explains that neurons are learning cosets (if $f$ is a prime factor) or approximate cosets (if $f$ is not prime factor). See Figure 6 in the revised paper, and Figures 17 and 18 in the appendix of the revised paper to see the intermediate computations that directly align with the CRT, which occur whether the network learned clocks or pizzas, and whether it learned cosets or approximate cosets.
>
> 2. We do not draw conclusions based solely on our specific experimental settings. Our theoretical framework extends beyond specific hyperparameter settings, as evidenced by its ability to explain results across the existing literature -- results that quite literally can be different circuits at the microscopic scale (Zhong et al. [4]). All the previous settings from the literature are generalized by our approximate Chinese Remainder Theorem model giving the bound of $\mathcal{O}(\log(n))$ in sections 5.2 and 5.4. Our mathematical model matches the behaviour described by every paper we build upon, namely, Nanda et al. [2], Chughtai et al., Gromov and Zhong et al., with the added benefit of finding a coset algorithm as Stander et al. did, thus reopening the universality hypothesis on finite groups, previously closed by Zhong et al's result that clocks, or pizza algorithms can be discovered (giving no universality on even just one finite group, nevermind possibly all of them). Finally, it is essential to mention that Nanda did vast hyperparameter explorations, even across depth and architectures, and always found that the number of frequencies was on the order of $\mathcal{O}(\log(n))$. Thus our results are backed by substantially more evidence than just our experiments. To alleviate your concerns however, we add Figure 29 and 30 to the appendix showing that the $\mathcal{O}(\log(n))$ is on the correct order as we scale the number of neurons and also as we scale $n$, the latter no preceding work has scaled on this problem. We also took Zhong et al.'s exact pizza network which they trained on mod 59 and show in Figure 17 and Figure 18, in the revised submission, that it is qualitatively implementing the approximate CRT we outline.

---

> > ### Author Response · Authors · 2024-11-21
> > **Official Comment by Authors**
> >
> > **Questions:**
> > 1. Section 5.2 is not a proof, but is a derivation. It shows how a neuron "knows" what a coset or approximate coset is. We have made the notion of approximate coset more precise in Section 3 so it comes well before its usage in section 5. Thank you for the suggestion as well as for pointing out the typo, which we fixed.
> > 2. The beginning of 5.3 was supposed to address Chughtai et al. [1], as rotation matrices (the representation matrices for the cyclic group) fail to explain the phase shifts and the values outside the (-1, 1) interval found in the embedding histogram. It has been corrected and now makes sense, thanks for catching this.
> > 3. Please note the signed distance \{ \} definition on lines 408-409 of the original submission. If $k$ is close to an integer then $\{k\}$ is close to $0$.
> > 4. Yes, the conditional frequencies will of course better minimize the loss if they are frequencies that naturally complement each other and intersect in a way such that they perform the CRT. This point being that it's best if the frequencies don't "happen" to intersect on more than just the correct logit. Consider the following example: if logit $k$ is the correct logit, all clusters (cosets) output large positive values on $k$, and clusters will also output large positive values on different $k' \not = k$, based on the frequency of the cluster. Both $\frac{f}{2}$ and $2f$ are going to increase the value of the correct logit $k$, but also end up intersecting and further boosting incorrect logits $k'$ that were already increased by the coset with frequency $f$ (because they intersect on many more logits than just the one correct logit). This would not be favourable as it would increase the loss, due to multiple wrong logits getting their values boosted to be more positive from two clusters. See Figure 5 in the original, now Figure 6 to better understand this. This figure should help you imagine what would happen if a cluster had $2f$ -- it would cause many logits to be larger in the total contributions plot (the bottom-most plot).
> >
> > [1] Stander et al., "Grokking Group Multiplication with Cosets" (2023)
> >
> > [2] Nanda et al., "Progress measures for grokking via mechanistic interpretability" (2023)
> >
> > [3] Gromov, "Grokking Modular Arithmetic" (2023)
> >
> > [4] Zhong et al., "The Clock and the Pizza: Two Stories in Mechanistic Explanation of Neural Networks" (2023)

---

> > ### Comment · Reviewer_qgqV · 2024-11-25
> >
> > I would like to thank the authors for offering new experiments and discussions. I have updated my score accordingly.

---

> > > ### Author Response · Authors · 2024-11-25
> > > **Official Comment by Authors**
> > >
> > > Thank you for your updated feedback and for acknowledging the additional experiments and discussions we provided. We sincerely appreciate the time you’ve taken to engage with our work.
> > >
> > > Could you kindly confirm if our responses have addressed your concerns regarding the novelty of the paper? If there are remaining concerns that lead you to consider a marginal rejection, we would greatly appreciate your clarification. Understanding these points would help us better address them and improve the work.
> > >
> > > We would also welcome any further questions or thoughts that might arise from discussions with other reviewers. Thank you again for your time and consideration, and we look forward to continuing the dialogue.

---

### Official Review · Reviewer_Km3r · 2024-11-02

**Soundness:** 3
**Presentation:** 3
**Contribution:** 2
**Rating:** 5
**Confidence:** 4

**Summary:**

The authors revisit the problem of reverse engineering neural networks trained to learn modular addition, attempting to unify seemingly different algorithms learned by neural networks in the literature. In particular, the authors identify “simple” neurons and “fine-tuning” neurons, and build a mathematical model for them. First, the authors explain that simple neurons implement approximate coset. The authors also demonstrate that the constructive interference of multiple frequency components leads to maximum signal (logits), and identify its relation to the Chinese Remainder Theorem (CRT) when the modulus is composite.

**Strengths:**

- The paper explores reverse engineering neural networks trained to learn composite modulus, which has rarely been studied in the literature.
- The paper develops an interesting mathematical model that can account for seemingly different learned algorithms in the literature.
- The paper effectively links the mathematical model with experimental results.

**Weaknesses:**

While this paper presents a lot of interesting ideas and results, I believe that further analysis on (a) the identified frequency clusters and (b) networks trained to learn composite modular addition would significantly strengthen its contributions.

The formalization of constructive interference between different frequency components, which was briefly introduced by Pearce et al. (2023), is a valuable aspect of this work. However, I feel that certain analyses are missing. For example:

- According to authors’ analysis, the number of frequency clusters depends on $O(log n)$. It would be valuable to experimentally validate this for a broader range of $n$, say from 3 to 997, not just around 90.
- Do all neurons in the same cluster play an equally important role in computing the answer?
- Does the model performance significantly drop if we ablate any of the $O(log n)$ clusters? What happens if we ablate only a part of the cluster? Models may develop unimportant frequency clusters that are not crucial for computation.
- What exactly are the chosen frequencies? Are they random, or do they satisfy certain constraints? For instance, in Figure 4(b), it would be interesting to include a plot that shows which frequencies tend to co-occur given a random seed.

For learning composite modulo addition, the authors explain that the constructive interference is very much analogous to CRT; Providing additional evidence that neural networks indeed implement CRT - such as by localizing intermediate computation results - would further bolster the paper’s contribution.

**Questions:**

- Typo in Line 330 ($n\rightarrow n'$ ). I think it may be clearer to reorganize section 5.2 in terms of theorems, i.e., first provide a formal definition of “approximate coset,” and then provide a proof.
- At the beginning of 5.3, I think neurons in the same cluster having different phase does not necessarily disprove Nanda et al. (2023a)’s algorithm - It is possible that there are multiple circuits doing the same operation (starting by translating one-hot $a,b$ into Fourier basis) within the model, each corresponding to neurons of same frequency but different phase.
- It is unclear if the approximation in Line 412 is valid. $cos(2\pi k) \approx 1-(2\pi k)^2/2$ is a valid approximation when $k\ll1$, not when $k$ is close to integers.
- Do you have any explanations for the conditional distribution of frequencies, on why it is less likely to learn $2f$ or $f/2$ when $f$ is learned?

---

> ### Author Response · Authors · 2024-11-21
> **Official Comment by Authors**
>
> Thank you for your review!
>
> **Weaknesses:**
>
> 1. Our bound of $\mathcal{O}(\log(n))$ matches all known experimental results in the literature. That said, we added experiments to the appendix (see Figure 29 and Figure 30 in the revised paper) to see that the bound is still satisfied as the number of neurons is scaled, and as the modulus $n$ is scaled. Also the $\mathcal{O}(\log(n))$ bound matches the Chinese Remainder Theorem (CRT), which decomposes a number into $\mathcal{O}(\log(n))$ factors. It is therefore unlikely that it can be beaten, and should be a topic of future work.
>
> 2. No, not all neurons play the same role, some have larger activations than others, and therefore will contribute more to the correct logit. For the most part, however, all of the activations are very close to each other and there isn't too much variance within an architecture, *i.e.* all models with 128 neurons learn neurons with preactivations in similar ranges, similarly for 512, 2048, 8096, etc.
>
> 3. Please see Figure 27 in the Appendix of the original submission for ablations, Figure 26 may also be of interest, due to it showing how injecting noise into all the weights of neurons in just one cluster affects the output. All frequency clusters are crucial for computation in the ablations we performed, *i.e.* we did not see redundant frequency clusters being learned.
>
> 4. Figure 6 in the revised paper (originally Figure 5) has been made more clear in order to show that the frequencies corresponding to clusters within a random seed are 35, 25, 8, 42, in descending order in the figure. The distributions of frequencies were provided because they show strong support for CRT being the algorithm (and they were indeed the initial insight that made us suspect this). The fact that primes and composites have the same frequency distribution was the first thing to make us suspect this, as we did not expect them to be the same when we started this research. We provided the conditional frequency distribution because it was not what you might expect at first glance, given that the overall frequency distribution is uniform.
> 5. We do localize intermediate computation results, please see Fig. 5 in the original submission (now Fig. 6). This figure is showing the approximate CRT algorithm. Each cluster is outputting values on a coset, and the logit that is green in all of the clusters is the logit that has its final output value boosted to be substantially larger than the next largest value. This is done via linear combinations of cosets, but it is effectively a set intersection operation on cosets, as the argmax will select the largest logit, and the largest logit is created by being one of the largest members of each coset. Thus the neural network is converging to an approximate CRT. Also, please see Appendix B.1.1, Figure 17 and Figure 18 (all in the revised paper), which show no clear difference at the abstract level between pizzas and clocks, achieved by localizing intermediate computation results.

---

> > ### Author Response · Authors · 2024-11-21
> > **Official Comment by Authors**
> >
> > **Questions:**
> > 1. Section 5.2 is not a proof, but is a derivation. It shows how a neuron "knows" what a coset or approximate coset is. We have made the notion of approximate coset more precise in Section 3 so it comes well before its usage in section 5. Thank you for the suggestion as well as for pointing out the typo, which we fixed.
> > 2. The beginning of 5.3 was supposed to address Chughtai et al. [1], as rotation matrices (the representation matrices for the cyclic group) fail to explain the phase shifts and the values outside the (-1, 1) interval found in the embedding histogram. It has been corrected and now makes sense, thanks for catching this.
> > 3. Please note the signed distance \{ \} definition on lines 408-409 of the original submission. If $k$ is close to an integer then $\{k\}$ is close to $0$.
> > 4. Yes, the conditional frequencies will of course better minimize the loss if they are frequencies that naturally complement each other and intersect in a way such that they perform the CRT. This point being that it's best if the frequencies don't "happen" to intersect on more than just the correct logit. Consider the following example: if logit $k$ is the correct logit, all clusters (cosets) output large positive values on $k$, and clusters will also output large positive values on different $k' \not = k$, based on the frequency of the cluster. Both $\frac{f}{2}$ and $2f$ are going to increase the value of the correct logit $k$, but also end up intersecting and further boosting incorrect logits $k'$ that were already increased by the coset with frequency $f$ (because they intersect on many more logits than just the one correct logit). This would not be favourable as it would increase the loss, due to multiple wrong logits getting their values boosted to be more positive from two clusters. See Figure 5 in the original, now Figure 6 to better understand this. This figure should help you imagine what would happen if a cluster had $2f$ -- it would cause many logits to be larger in the total contributions plot (the bottom-most plot).
> >
> > [1] Chughtai et al., "A Toy Model of Universality: Reverse Engineering How Networks Learn Group Operations" (2023)

---

> > > ### Comment · Reviewer_Km3r · 2024-11-24
> > >
> > > Thank you so much for your comments and detailed revisions! Overall, I’m quite excited about the paper—it presents interesting observations and an attempt to unify seemingly different algorithms learned by neural network. I also found the idea that neural networks may implement CRT-style algorithms fascinating, although I feel the direct evidence supporting it (e.g., a CRT circuit) is currently missing. I’ve adjusted my score accordingly.

---

> ### Author Response · Authors · 2024-11-25
> **Official Comment by Authors**
>
> Thank you for your comments and positive feedback! We are also excited by these developments.
>
> We show the network's implementation of the Chinese Remainder Theorem (CRT) algorithm in Figure 6 and Figures 17 and 31 in the appendix (in the revised version of the paper).
>
> The reason that this is a "CRT" algorithm is because of how similar it is to the CRT. We will make this precise below:
>
> Suppose that you are doing addition mod 105, then there are 3 prime factors: 3,5,7. Suppose the answer to $(a+b)\mod105 = 10$.
>
> You can use cosets (the CRT) in the following way:
>
> Compute $10 \mod 3 = 1$. Now get the coset of values mod 3 = 1, i.e. {1, 4, 7, **10**, 13, ...}
>
> Compute $10 \mod 5 = 0$. Now get the coset of values mod 5 = 0, i.e. {0, 5, **10**, 15, ...}
>
> Compute $10 \mod 7 = 3$. Now get the coset of values mod 7 = 3, i.e. {3, **10**, 17, 24, ...}
>
> Now the intersection of these three sets is 10, i.e. it is the only number that is in all three sets. To turn this into what the neural network has "learned", we have to do the following:
>
> 1: fit a sine wave through these cosets with a frequency such that it peaks on each value in the coset (in this case that frequency is the prime factor).
>
> 2: fit a sine wave through every coset we did not mention, e.g. (mod 3 = 0, mod 3 = 2, etc).
>
> Now, when we cluster all sine waves (neurons) with the same frequency, we get three clusters and the behavior shown in Figures 6, 17 and 31 *i.e.* the approx CRT! Furthermore, we also get the behavior shown in 31, namely, when you shift (a, b) both by two, all the logit values that come from each cluster are shifted in a periodic way (the entire picture just shifts by 4 to the right). This is actually equivariant behavior! Each cluster has indeed learned all the cosets of a particular type, and when a or b is incremented, the cluster shifts its output appropriately.
>
> This is the approximate CRT. The neural network has effectively implemented the set intersection operation by adding linear combinations of cosets together. Naturally, only one logit is in the intersection, so only one logit gets a large output value! Brilliantly, the network learns frequencies that naturally complement each other. We show this in our histograms of learned frequencies (conditional frequencies) in Figure 4b. The network is much less likely to learn frequencies that are multiplicatively related because these frequencies would "intersect" on more than one value, i.e. the correct logit would be in the intersection, but so would other logits (between any clusters with multiplicatively related frequencies). This is further quantitative evidence for the approximate CRT algorithm.
>
> We have added this explanation directly under remark 1 in the appendix.
>
> The last thing we'd like to say for now, is that the above procedure is invariant throughout all interpretations in the literature. The various interpretations give different low-level implementations of this approximate CRT procedure described above. Despite the interpretations being substantially different at the low-level, they all implement clusters of cosets, that are then linearly combined in such a way that the intersection has its value maximized with a large separation between the correct logit and second highest valued logit. Furthermore, all of the empirical results in the literature use around O(log(n)) frequencies, though they do not include the number of frequencies to be part of their interpretation. This is exactly what our model predicts! This aligns well with the CRT, which also uses O(log(n)) factors for solving a modular addition problem. These O(log(n)) factors come from the fact that a number can not have more than O(log(n)) factors. Thus, our interpretation (that networks implement the approximate CRT), is backed by:
> 1) a mathematical model predicting log(n) frequencies
> 2) experimental results showing log(n) frequencies are always the number found in reverse engineered networks
> 3) Figures 6, 17, 31 all showing that the network has implemented the CRT
> 4) the conditional frequencies distribution shows that the network tries to avoid multiplicatively related frequencies, i.e. it avoids $2f$ and $\frac{f}{2}$, which indeed would have many logits in their intersections, and would thus not be "ideal" frequencies to pair with $f$, due to not sufficiently lowering the loss.
>
> Considering this construction for how to build a CRT out of periodic functions (a construction that aligns with all experimental results), along with Figures 6, 17, 31, and the added example to remark 1 in the appendix, we hope this alleviates your concerns regarding the absence of direct evidence for the CRT. If this addresses your concerns, we kindly invite you to reconsider your recommendation to reject the paper. However, if further clarification is needed, we are more than willing to continue the discussion and to produce any additional figures or analyses that you believe would more effectively demonstrate the CRT.

---

> > ### Author Response · Authors · 2024-12-02
> > **Response by the authors**
> >
> > We moved the CRT algorithm from the appendix to the main text, and added a description about how it works in section 3. We hope this clarifies how the CRT algorithm is implemented by the network. Please also see the new scaling plot: https://postimg.cc/0K973RqB. Our mathematical model predicting that networks will implement the CRT with $\mathcal{O}(log(n))$ clusters of frequencies is now strongly supported by experimental evidence.
> >
> > Thank you for your comments, especially that you wanted the CRT circuit explained in the main text. We believe the revision has greatly benefited from its inclusion.

---

### Official Review · Reviewer_ZW9m · 2024-11-03

**Soundness:** 2
**Presentation:** 1
**Contribution:** 2
**Rating:** 3
**Confidence:** 3

**Summary:**

The premise of this paper is to understand how modular representations appear in networks and introduces language to characterize them (i.e. coset structures/circuits, "simple" neurons, and "fine-tuning" neurons). This paper also offers theory for identifying these structures in trained networks.

**Strengths:**

- In general, this paper is interesting in the way that it conveys notions of model superposition and modularity with group-theoretic ideas.

- Fig. 3b is nice in that it offers some intuition relating "simple" neurons and coset structures.

- "Re-opened conjecture 2" is certainly interesting.

**Weaknesses:**

$\underline{\text{Nits}}$

L298: modelling $\to$ modeling

Figure 5: Why are "Contributions" and "Clusters" capitalized?

Frequent use of the LaTeX \citet function instead of \citep.

Figure axis labels and titles are extremely difficult to parse/understand. Figures themselves are poorly designed and constructed.

$\underline{\text{General}}$

While the ideas in this paper are interesting and offer some new perspectives on well-studied topics in ML literature, I believe this paper is somewhat poorly written, making it difficult to properly assess.

- What is Fig. 1 doing on page 2? I had a difficult time understanding in how this figure lends to the story being described in the introduction, only to find it finally referenced a few pages later. Even then, the figure captions and titles did not help me understand the figures easily.

- Many of the analyses are not well motivated. For instance, the first section of the Experimental Findings immediately delves into understanding/characterizing the difference in networks trained with batch sizes = n and with batch sizes = size of the full training dataset. Why? What motivates this analysis? Although I found much later in the paper that this was something studied in one of the reference papers, this paper needs a major revision/reworking in order to convey the story better. Fig. 2 can be a lot clearer in explaining why this analysis is necessary. Finally, what do you expect to happen for weights when trained on SGD (batch sizes = n) vs GD (batch sizes = size of dataset)? Yes, you mention that smaller batches find sparser embeddings. But what does this information provide the reader?

- If the goal is to show that across 100k trained networks (of varying seed) that the coset structures generally emerge, wouldn't one also do strict ablations / variations of learning rate, model architectures (larger/smaller parameters, varying depth), choice of optimizer, and with and without the L1 regularization?

- Although I like Fig. 3b and the accompanying text that describes these findings, how significant is this result? Would $\textit{any}$ network trained on $\textit{any}$ inherently modular task learn redundant functions/circuits/representations? Additionally, this network is a simple 1-hidden layer MLP. How do your conclusions hold for MLPs of varying depth?

- L344-352: I am unsure if this text is proposing this information as a novel contribution of this paper. Model superposition papers exist in plethora which show that orthogonality of learned features/functions are not optimal in many cases, and that smaller networks replicate the circuitry of larger, more sparse networks in specific cases.

- Do you really need L1 regularization in Conjecture 1? Additionally, wouldn't Conjecture 1 only hold for MLPs of 1 hidden layer?

- A general comment I have is that some intuitions for how this group-theoretic perspective can aid in understanding modularity and model superposition in networks trained on non-mathematical tasks would be especially helpful, considering since these topics are well studied in standard ML / mech. interp. literature.

**Questions:**

See above.

---

> ### Author Response · Authors · 2024-11-21
> **Official Comment by Authors**
>
> Thank you for your review!
>
> 1. Thank you for pointing this out--we would be happy to move the figure, was there any particular part of the text that you would have expected it to fit more naturally? We are open to suggestions on presentation. As is, we plan to improve the quality of figures as has already been done for Figure 6.
>
> 2. We appreciate the reviewer's comments about motivation and would like to clarify the progression of our analysis. The comparison of different batch sizes builds directly on a fundamental question in the field: what mathematical structures do neural networks learn when solving modular arithmetic? Prior work (e.g. Chughtai et al. [4]) proposed that networks learn rotation matrices on modular arithmetic, which would necessarily have values bounded between (-1, 1). Our Figure 3 (originally Figure 2) serves a crucial purpose in testing this hypothesis -- it reveals that networks trained with small batch sizes, compared to the full batch size settings studied in Chughtai et al., learn embeddings with values outside this range, suggesting the underlying mathematical structure may be more general than Chughtai et. al. claim. The batch size comparison thus provides key evidence for motivating section 5.3 on projections of representations. Regarding the sparse embeddings observation, we can remove this remark if you feel it detracts from the paper's clarity. It is not a point of the paper and is only an observation.
>
> 3. Our theoretical framework extends beyond specific hyperparameter settings, as evidenced by its ability to explain results across the existing literature. The works of Nanda et al. [3] and Chughtai et al. have already extensively explored different architectures and hyperparameter settings, and all their experimental results conform to our $\mathcal{O}(\log(n))$ bound. Similarly, Zhong et al.'s results, obtained under different conditions, align with our predictions. Gromov's [5] theoretical work, using $\mathcal{O}(n)$ frequencies and due to neurons with random phases, cannot yield cosets and approximate Chinese Remainder Theorem (CRT). This consistency across diverse experimental conditions suggests that our findings capture fundamental mathematical structures rather than artifacts of specific hyperparameter choices. Additionally, we use L2 regularization, not L1, in line with existing literature (e.g. [1]), as it has been established as the appropriate inductive bias for this problem since the findings of the original grokking paper [2].
>
> 4. Figure 4 (the histograms of learned frequencies, originally Figure 3), is significant because the fact that it's uniform gives a ton of insight into what's actually going on. By having no preference for any frequencies, it shows that group isomorphisms are at play, and indeed we use this fact to remap both simple and fine-tuning neurons to frequencies of 1 and 1/2, respectively. Doing so allows for a qualitative difference between simple and fine-tuning neurons to be made, in addition to the quantitative difference provided by the differing DFTs. The conditional frequency histogram Figure 4b (originally 3b) is less significant, and is shown in order to point out that the conditional distribution of learned frequencies is more complicated than you'd expect given that the distribution of frequencies is uniform. Regarding network depth and architecture, our conclusions are supported by Nanda et al.'s extensive experiments across different MLP depths and transformer architectures, all of which exhibit the $\mathcal{O}(\log(n))$ frequencies scaling we predict. Regarding the broader question of modular tasks, we find no evidence that networks learn more redundant circuits for composite moduli compared to prime moduli. This is actually a key finding because we first saw approximate CRT on composites, and then tested primes and found to our surprise the exact same algorithm, despite primes having no cosets, only approximate cosets.
>
> 5. L344-352: This is a novel contribution of this paper. No preceding paper has come close to a $\mathcal{O}(\log(n))$ bound on this problem, the current record is by Gromov and is $\mathcal{O}(n)$, which used superposition and a totally different mathematical approach (though he modeled single neurons a similar way). Furthermore, no preceding paper has proposed the approximate CRT as an interpretation of what neural networks learn for this problem.

---

> > ### Author Response · Authors · 2024-11-21
> > **Official Comment by Authors**
> >
> > 6. L2 regularization may not be needed in the conjecture, but is included due to this dataset being very hard in even the medium data regime, nevermind being impossible in the low data regime without L2 regularization. No, conjecture 1 should hold for multilayer MLPs and Transformers. Evidence for this is that Nanda et al. trained a 2-layer Transformers and inspection of his results yields that the number of key frequencies found were on the order of $\mathcal{O}(log(n))$. Neurons adhering to these frequencies can be clustered, and there will thus be $\mathcal{O}(log(n))$ clusters, each cluster forming an approximate coset. As depth is increased, clustering may become more involved.
> >
> > 7. To address your general comment, we refer to our global rebuttal. It's primarily through the universality hypothesis that this work relates to standard ML mechanistic interpretability literature, but it also relates to it because we show that all of the "conflicting" interpretations are actually the same when viewed abstractly. This fact has implications to all interpretability research, namely "what makes an interpretation good"? We wrap up and connect the five preceding interpretability papers that were previously disconnected. To emphasize this point about abstractions, we add "Thus, it is the case that the results found by Nanda et al. (2023a); Chughtai et al. (2023b); Zhong et al. (2024) are all true in their own right, yet simultaneously conflict for being what the neural network learned. This is because the explanations are not robust at multiple scales, giving rise to an interesting question: should we consider an interpretation to be good only if it’s true at more than one scale of abstraction?" to Section 6 to better reflect this. Furthermore, our bound of  $\mathcal{O}(log(n))$ is the best in the literature for this problem, beating the next best by an exponential amount as it is $\mathcal{O}(n)$ in Gromov's work.
> >
> > **Suggestions:** Thanks for your suggestions and for pointing out typos in the paper. We are planning to modify figures. So far, we already greatly improved Figure 6 (originally Figure 5). We welcome any further suggestions.
> >
> > We hope this provides sufficient clarification, and we are open to further discussion about how this paper fits into the current literature as well as answer any more questions.
> >
> > [1] Liu et al., "Omnigrok: Grokking Beyond Algorithmic Data" (2022)
> >
> > [2] Power et al., "Grokking: Generalization Beyond Overfitting on Small Algorithmic Datasets" (2022)
> >
> > [3] Nanda et al., "Progress measures for grokking via mechanistic interpretability" (2023)
> >
> > [4] Chughtai et al., "A Toy Model of Universality: Reverse Engineering How Networks Learn Group Operations" (2023)
> >
> > [5] Gromov, "Grokking modular arithmetic" (2023)

---

> > > ### Comment · Reviewer_ZW9m · 2024-11-25
> > > **Response to Rebuttal**
> > >
> > > Thank you for your response.
> > >
> > > 1. This updated Figure (Figure 2 in your new PDF, Figure 1 in your old submission) should appear where it is first referenced. As far as I can tell, it is referenced somewhere in Section 5.5. Additionally, your Figure 1 is never referenced in text. I understand what you might be trying to convey, but this figure, along with many others, are just never referenced or explained. It feels like there's been an overwhelming dump of new figures in this revision without too much analysis or careful reorganization of thoughts.
> > >
> > > 2. Thank you for this comment. This motivation was totally unclear in reading the paper because there was no clear progression of thoughts or motivation for analyses. Although I appreciate your small caption change in the paper, this is still unclear in this revision. Additionally, why can't one say that the circuits you find now are just scaled or transformed versions of rotation matrices? It feels like these analysis steps that should be motivating your main science need more rigor.
> > >
> > > 3-5. I think these explanations should be made clearer in the text.
> > >
> > > 6. Got it, thank you for this point.
> > >
> > > 7. I think this has become clearer from seeing your new Figure 1 and totally new abstract.
> > >
> > > In general, I thank the authors for their detailed revisions. While there have been some clarifications and changes in text that have reduced _some_ confusion on my part, I believe that this new revision of small changes in text and some 15+ new figures has generally not reduced _most_ of the confusion I had in reading this paper. It feels like there's little analysis of these new plots in the main text, and whatever analysis motivates your main science is lacking rigor (I would suggest toning down the strength of some claims made in these motivating sections).

---

### Official Review · Reviewer_cGQM · 2024-11-05

**Soundness:** 3
**Presentation:** 2
**Contribution:** 2
**Rating:** 5
**Confidence:** 4

**Summary:**

The paper studies modular addition (prime and composite) with a 2-layer ReLU-MLP. They interpret the learned algorithm in the language of cosets (and the so-called approximate cosets) in conjunction with the known Fourier features. They present empirical evidence for their suggested algorithm. Using their findings, they attempt to unify the findings of prior works on the topic.

**Strengths:**

- Studying composite modulus for modular addition and highlighting the subtlety with cosets is new.

- I find the connection to non-circular algorithms found in [1] (with Lissajous curves) to be quite interesting (Lines 477-481).

- In networks exhibiting frequency clusters (such as [2]), discussing the phase shifts within a frequency-cluster is new, to my knowledge.

- I appreciate the authors averaging their experiments of 100k random seeds to eliminate fluctuations.

**Weaknesses:**

My main concern about this work lies in its novelty/contribution.

- For prime moduli, the current work offers very little more insight than the union of prior works [2, 3]. Specifically, the superposition of multiple frequencies resulting in high output at the correct logit has been extensively studied in [3]. The current paper studies a slightly different setup, which results in frequency clusters (like in [2]) rather than individual frequency neurons (like in [3]). However, the paper effectively distills down to a superposition of these works. While the result itself is new in this setting, the insight gained is limited.

- For prime moduli, the coset picture does not add any more insight, since all cosets are of order 1 ($gcd(f,n)=1 \\;\\forall f$). For composite moduli, one would expect that having multiple elements in coset would affect the prediction accuracy of individual neurons. However, Fig. 4 shows no difference between $n=89$ (prime) and $n=91$ (composite). Furthermore, the "approximate coset" interpretation is just a different way to describe the algorithm found in [3], to my understanding.

Additionally, some of the claims made in the paper are not presented with sufficient empirical evidence (Elaborated in the Questions).

**Questions:**

- How do the magnitude histograms shown in Fig. 2 prove that rotation matrices are not learnt? I think this claim requires further justification.

- Lines 208-211: How is this finding different from the results from [3]?

- Line 232: Isn't it more reasonable to look at relative magnitude of highest frequency (relative to other frequencies within the neuron) rather than absolute magnitude?

- Section 5.3: It is unclear how the suggested representations are drastically different from the rotation matrices. Aren't they just shifted+rescaled rotation matrices?

- Lines 450-456: The result that the number of clusters $\\approx log_e n$ requires further verification. One needs to empirically verify this using multiple different moduli -- and check if it scales correctly. For one/two moduli, the values of $\\delta$ and $\\rho$ can be cherry-picked to match the observed number of clusters.

- Lines 477-481: This is an interesting hypothesis. But it needs to be made more precise and shown empirically.

- It is unclear to me how the presented algorithm unifies clock and pizza algorithms from [1]. Could the authors clarify this?

Suggestions:

- Line 241: "Approximate coset" term is used here, but defined much later -- this hurts the clarity.

- Line 330: potential typo: "multiple of $n'$" instead of "multiple of $n$"

- Lines 327-336: The explanation for approximate coset might benefit from a simple figure.

[1] Zhong et al., "The Clock and the Pizza: Two Stories in Mechanistic Explanation of Neural Networks" (2023)

[2] Nanda et al., "Progress measures of grokking via mechanistic interpretability" (2023)

[3] Gromov, "Grokking Modular Arithmetic" (2023)

---

> ### Author Response · Authors · 2024-11-21
> **Official Comment by Authors**
>
> Thank you for your thoughtful review!
>
>
> **Weaknesses:**
>
> Our global comments address the novelty and contribution, but we are happy to answer any other questions.
>
> 1. The insight gained here is not limited when you consider that we re-open the universality hypothesis and successfully unify all the existing literature. Our abstraction of cosets and Chinese Remainder Theorem (CRT) fits and tightly bounds the empirically-observed algorithms of Nanda et al., Chughtai et al., Zhong et al. all under one general algorithm, being CRT on cosets and approximate cosets. None of these prior works identified (or mentioned) that their algorithms were low-level implementations of a more abstract coset algorithm that was being learned by the network. Note Gromov's paper required $\frac{n-1}{2}$, *i.e.* $\mathcal{O}(n)$ frequencies and is effectively a brute-force model that doesn't match experimental results, thus it is not generalized by our solution, which is a much tighter model giving  $\mathcal{O}(\log(n))$.
>
> 2. Please consider the fact that the neural network is always qualitatively and quantitatively *the same*, as you point out "Fig. 4 shows no difference" regardless of whether it is trained on a prime or composite modulus. This is true even when it learns exact cosets for a composite and (always) learns approximate cosets for a prime. This was surprising to us, when we started this research we expected a difference. The finding that there was no difference was what gave us the insight to realize that the abstract algorithm the neural network is always converging to is an approximate CRT -- even when there are no cosets! In the prime case of no cosets, it uses approximate cosets, and intersects them exactly like they're cosets. The finding that it uses approximate cosets just like it uses cosets in order to do an approximate CRT is a significant insight into the prime number case. It is in fact the insight that let us generalize to everything else, because it led us to suspect and eventually identify the approximate CRT algorithm. Furthermore, the approximate CRT algorithm is *not* another way to describe the algorithm in Gromov, as it uses exponentially fewer frequencies.

---

> > ### Author Response · Authors · 2024-11-21
> > **Official Comment by Authors**
> >
> > **Questions:**
> >
> > 1. (**and question 4.**) The rotation matrix values are all bounded by $(-1, 1)$ and the fact that a large percentage of values $(\sim 5$%) are showing up outside this range implies that it may not be rotation matrices. Further evidence against rotation matrices is that they would not explain the different phase shifts within the clusters. Thus, we pose scaled projections of representations as a more general model. **4.** Our paper states, "Note that such structure is implied by the hypothesis that neural networks
> > trained on group tasks learn representations, but is more general because of the existence of both amplitude and phase shifts." Projections of representations are implied by Chughthai et. al.'s representations hypothesis and thus are not drastically different, but just slightly more general. Also note that Chughtai et. al. propose that the solutions discovered have as weights components of linear representation matrices, which are indeed rotation matrices, but a very specific type of rotation that have a period that divides the order of the group. Our experiments show this is not the case in general. Supporting projections of representations as a better model.
> >
> > 2. 208-211: Gromov used quadratic activations and modeled what we're calling simple neurons. This finding is a replication of Gromov, we will add a citation, but lines 211-215 deviate from Gromov as he did not mention or remark about anything resembling a fine-tuning neuron. It's worth noting that Gromov gives an exact solution for quadratic activations requiring $\mathcal{O}(n)$ frequencies, whereas we give a $\mathcal{O}\log(n)$ frequencies algorithm with ReLU activations and point out that the evidence implying the universality hypothesis was false doesn't make it false, and thus universality is still an open problem on finite groups at an abstract algorithm-level. That said, we have realized it is necessary that we add mentions of Gromov to multiple places and have done so.
> >
> > 3. 232: Yes, it is more reasonable to look at the relative magnitude. For our new plots for 512, 2048 and 8096 neurons we use relative magnitudes of $0.06 \cdot$max\_frequency\_magnitude to detect fine-tuning neurons. It's worth noting that this is only to filter out fine-tuning neurons vs. simple neurons. Were this to miss, it would not change the number of clusters detected.
> >
> > 4. This is answered in question 1.
> >
> > 5. Agreed, we believe this is a very interesting and important direction. We estimate empirically that $log(n)$ tends to be an upper bound over the range that we can test (see Fig 29 and Fig 30 in the revised appendix for scaling experiments), but further investigation would be impactful. We'd like to point out that all reported experimental results in the literature, which use various hyperparameters, architectures, and values for the modulus (e.g. Zhong et al. use 59, Pearce et al. use 67, Nanda et al. use 113), all fit under a $\mathcal{O}(\log(n))$ bound for the number of required frequencies.
> >
> > 6. This claim follows from our mathematical model, but we added experimental evidence that perfectly matches this prediction by the model (see appendix B.1). We do this by replicating Zhong et al. and simply adding a DFT to their principal component (PC) plots. This clearly shows that when PCs come from different clusters, the non-circular embeddings are formed. Thank you for suggesting we verify this. The fact our model predicted it and it was easy to experimentally verify strengthens the paper.
> >
> > 7. Both the clock and the pizza algorithm operate on cosets in Zhong et al. Their Figure 1 shows the pizza algorithm to be clustering things together based on coset membership. Where the figure says "Same-label predictions are spread over two slices of pizza", the authors list a set of pairs $(a,b)$ satisfying $(a+b) \equiv 5 \mod 12$. In modular arithmetic, cosets arise when we consider equivalence classes defined by an equation like this. We make this clear in B.1.1 Figure 17 in our revised paper, by showing that there is no obvious qualitative or quantitative differences between the approximate CRT algorithm when it is carried out by pizzas or clocks. Note, for example, that the output logits for the cluster with max frequency equal to 15 has maximum activation along an approximate coset $\frac{59}{4}=3.93$, and if the neuron activates strongly at $a$ then it also activates strongly at $a\pm3.93\sim 4$.
> >
> >
> > **Suggestions:** Thank you for the suggestions! We included a definition of approximate cosets in section 3, fixed the typo and we plan to include a figure to illustrate approximate cosets in a future revision.

---

> > > ### Comment · Reviewer_cGQM · 2024-11-25
> > >
> > > I thank the authors for answering each of my questions and concerns in detail. I especially appreciate their efforts in adding new experiments and re-writing.
> > >
> > > ## $O(\\log n)$ frequency clusters
> > >
> > > I thank the authors for adding figures 29 and 30 in response to my question.
> > > - However, figure 30 is far from convincing of the author's original claim of $O(\\log n)$ clusters. In fact, even their modified claim of "upper bound" is vacuous given the evidence in figure 30. One can just as easily fit a constant or a linear (or virtually any other function) within the standard deviations shown in the figure. Even if one ignores the standard deviations, the average values do not show any inclination towards a logarithmic behaviour -- one can just as easily fit them with a linear curve of appropriate scale.
> > > - Another important point relates to the comparison with Gromov's work [1]. I disagree with the authors' comments that
> > > > (global comment) "mathematical models in the literature predict $O(n)$ frequencies are required (not matching experimental results)"
> > > >
> > > and
> > > > (line 122-123) "This model predicts $O(\\log n)$ frequencies (clusters), which matches experimental results in the literature, see Fig 5b, and improves on the previous model which uses $O(n)$ frequencies."
> > > >
> > > Gromov's model matches quite well with the experimental setup used in their work [1]. In fact, there are theoretical works that corroborate this, such as [2]. It is more accurate to say that Gromov's setup requires $O(n)$ frequencies without clusters, whereas the current setup (similar to some other works) requires **clusters** of fewer frequencies.
> > >
> > > The observation that a handful of frequencies are found by the model is of course known from prior works like Nanda's [3]. Thus, the attempted novelty of this work is the claim that the number of clusters in this setup follows (or is upper bounded by) $O(\\log n)$. However, the experimental evidence (figure 30) that this behaviour (or upper bound) is evident (non-vacuous) is not convincing.
> > >
> > > ## Newly added Section B.1
> > >
> > > While I commend the author's efforts in adding the new results in Section B.1, I think this section would benefit from a more descriptive and thoughtful writing.
> > > - It is not immediately clear what is meant by "DFT for PCx vs PCy" in the figure captions. Does it mean that the concatenated embeddings are projected onto the subspace spanned by the x-th and y-th principle components and then DFT is performed? Similarly, it is also not clear what is meant by "PCx (f=35) and PCy (f=8)". Are 35 and 8 the frequencies of the concatenated vectors projected onto the x-th and y-th principle components?
> > > - Furthermore, my suggestion for these experiments is to use a prime modulus to disentangle the artifacts of composite modulus from the main message, which is the relation between cosets and non-circular embeddings (The original paper [4] used prime modulus 59.)
> > > - Assuming my guess about the above definitions is correct, the only takeaway I can draw from figures 8-16 is that same frequencies in two PC directions lead to circles whereas different frequencies lead to non-circular Lissajous-like curves. I don't understand how this relates at all to cosets -- especially if one were to use a prime modulus. Can the authors clarify this further?
> > >
> > > ## Clock vs Pizza algorithm
> > >
> > > From figure 17 and the description around it, I understand that the authors are implicating that even in the Pizza case, the picture with constructive interference between the logits from various frequencies holds.
> > > - Can the authors explain, using their picture, what the difference is between the two algorithms? If pizza and clock are the same from their perspective, what explains the qualitative differences observed by the authors of [4]?
> > > - I think the authors should explicitly show that the model they use exhibits pizza algorithm -- or at least point to the exact setup and checkpoint that took from [4].
> > >
> > > [1] Gromov, "Grokking Modular Arithmetic" (2023)
> > >
> > > [2] Morwani et al., "Feature emergence via margin maximization: case studies in algebraic tasks" (2024)
> > >
> > > [3] Nanda et al., "Progress measures of grokking via mechanistic interpretability" (2023)
> > >
> > > [4] Zhong et al., "The Clock and the Pizza: Two Stories in Mechanistic Explanation of Neural Networks" (2023)

---

> > > > ### Comment · Reviewer_cGQM · 2024-11-26
> > > >
> > > > ## Zooming out
> > > >
> > > > From a broader perspective, I think that the authors have made some interesting observations and the work has potential value. However, the presentation of experiments and overall writing makes it difficult to verify the claims and appreciate the underlying coherent picture (for me, at least). In part, this may be due to the significant changes attempted during the short rebuttal window.
> > > >
> > > > To summarize my understanding of the current work: I agree with the idea that each frequency (neuron) selects a few values in the logits to be high; various frequency clusters and different phase-shifts within the clusters constructively interfere to select the correct output. (This takeaway has some conceptual similarity with Gromov's work, but some differences too, since this work uses a different setup that exhibits clusters of frequencies.) The authors' reformulation of this picture into the language of approximate cosets and CRT is also an interesting direction -- albeit, from the evidence presented the significance of this picture either lacks thorough verification or is made opaque behind the haphazard presentation.
> > > >
> > > > My suggestion to authors is to decouple the following two points:
> > > > 1. Use prime modulus for the purpose of unifying prior works -- such as clock/pizza/non-circular. Highlight with clear experiments and/or mathematical description the apparent differences in these algorithms and how they are actually different manifestations of the same underlying picture (aka CRT).
> > > > 2. Use composite moduli to further show the general applicability of the claims. Show the subtle differences arising from these non-prime moduli.
> > > >
> > > > ## Score
> > > > Due to aforementioned factors, I maintain my original assessment, which lies marginally below acceptance. That being said, I am confident that a thoughtful re-writing of the work will result in a valuable contribution.

---

> ### Author Response · Authors · 2024-12-02
> **Response by the authors**
>
> **$\mathcal{O}(\log(n))$ frequency clusters**
>
> > "However, figure 30 is far from convincing of the author's original claim of $\mathcal{O}(\log(n))$. clusters. In fact, even their modified claim of "upper bound" is vacuous given the evidence in figure 30. One can just as easily fit a constant or a linear (or virtually any other function) within the standard deviations shown in the figure. Even if one ignores the standard deviations, the average values do not show any inclination towards a logarithmic behaviour -- one can just as easily fit them with a linear curve of appropriate scale."
>
> Please see the new scaling plot: https://postimg.cc/0K973RqB. It scales out to 4999 and we check linear, square root, and logarithmic fits, getting $R^2$ values of:
>
> | Fit      | $R^2$      |
> |----------|----------|
> | Logarithmic  | 0.971  |
> | Square root  | 0.741  |
> | Linear  | 0.473  |
>
> We believe this is strong evidence for the $\mathcal{O}(log(n))$ bound, predicted by our theoretical CRT model.
>
> > Another important point relates to the comparison with Gromov's work [1]. I disagree with the authors' comments that (global comment) "mathematical models in the literature predict $\mathcal{O}(n)$ frequencies are required (not matching experimental results)"
>
> The results of Morwani et. al and Gromov assume no biases are in the network. Thus their experiments match their theoretical models for this different situation (no biases), and observe $\frac{n-1}{2}$ frequencies being learned. We study networks with biases, and provide a theoretical model predicting $\mathcal{O}(\log(n))$ frequency clusters, are required to get a constant separation between the correct logit output and second largest logit output. Our scaling plot now backs this experimentally. We hope this clarifies how our results are different to Morwani et al.'s and Gromov's -- both operate with no bias frameworks. Thus, we provide the first theoretical model that predicts what the network will learn in the more general case of training with biases.
>
> We did not originally realize that Gromov and Morwani et al., trained without biases. Thank you for challenging us on this point, because a great deal of the confusion in this discussion period was caused by it. We have addressed this difference in the revised paper and made it clear how we use biases and the two previous theoretical works do not.

---

> > ### Author Response · Authors · 2024-12-02
> > **Response by the authors**
> >
> > **Section B.1.**:
> > > It is not immediately clear what is meant by "DFT for PCx vs PCy" in the figure captions. Does it mean that the concatenated embeddings are projected onto the subspace spanned by the x-th and y-th principle components and then DFT is performed?
> >
> > Yes. It means that the two embedding **matrices** are concatenated, and then a PCA of this matrix is performed. The principal components are listed by order of contribution, *e.g.* PC1 is the largest contributing component, PC2 the second largest, etc. So PCx vs PCy can be for example Fig. 9 (PC1 vs PC2), showing both PCs coming from the same frequency -- proven by the DFT of the scatter plot detecting only one frequency of 35); thus they form a circular embedding, as predicted by our main text.  Thus these plots prove our claims in the text: circular embeddings are caused by two of the same frequency PCs being plotted, and non-circular embeddings are caused by two different PCs being plotted.
> >
> > > Similarly, it is also not clear what is meant by "PCx (f=35) and PCy (f=8)". Are 35 and 8 the frequencies of the concatenated vectors projected onto the x-th and y-th principle components?
> >
> > No. Each principal component corresponds to a specific frequency cluster. Please reference Figure 8, where we state which cluster has which frequency. These are the principal components coming from the concatenated embedding matrices, *i.e.* combining the embedding A and B matrices into one matrix, and then doing a PCA of the matrix. Not a PCA of specific input vectors. These results were added to support our claim in the main text, that Zhong et al.'s circular and non-circular embeddings are completely explained by simply noting that if the PCs come from the same cluster, it will be circular, else it will be non-circular.
> >
> > > Furthermore, my suggestion for these experiments is to use a prime modulus to disentangle the artifacts of composite modulus from the main message, which is the relation between cosets and non-circular embeddings (The original paper [4] used prime modulus 59.)
> >
> > One of the points of our paper is that the network is not doing anything qualitatively different after the embeddings for prime and composite moduli. This was incredibly surprising to us at first, because we assumed that it would be different for composite moduli. This means that the network may have neurons that output values on precise cosets if they learn a prime factor, but in the case of approximate cosets being learned, they are combined the exact same way as cosets would be combined. Prime or composite moduli -- the network processes inputs to outputs via the approximate CRT.
> >
> > We made all these plots also for a prime moduli, but since the only difference in these PCA plots is when the frequency the PC came from is a prime factor vs. not a prime factor, all possible combinations are already disentangled in Figs. 9 to 17. e.g. "Fig 9 shows two prime factors, showing a new type of circular embedding. Figs 10, 11 and 12 show a prime factor frequency vs. a non-factor frequency. Figs. 13-17 replicate the circular embeddings and Lissajous curves from Zhong et. al, because they are two non-factor frequencies simultaneously." It's easy to add the same plots for a prime moduli (we already have them), but we hope that with this explanation, you can see that giving the plots for composites includes all the information of primes plus more in a disentangled way.
> >
> > > Assuming my guess about the above definitions is correct, the only takeaway I can draw from figures 8-16 is that same frequencies in two PC directions lead to circles whereas different frequencies lead to non-circular Lissajous-like curves. I don't understand how this relates at all to cosets -- especially if one were to use a prime modulus. Can the authors clarify this further?
> >
> > Yes. The main take away from the PCAs was what you took away. See the next answer for an in depth explanation about approximate cosets.

---

> > > ### Author Response · Authors · 2024-12-02
> > > **Response by the authors**
> > >
> > > **Clock vs Pizza algorithm**
> > >
> > > > "Can the authors explain, using their picture, what the difference is between the two algorithms? If pizza and clock are the same from their perspective, what explains the qualitative differences observed by the authors of [4]?"
> > >
> > > They are different as observed by [4]. The point of our paper is that they are equivalent when coarse grained to be seen as outputting on approximate cosets. The qualitative difference is hidden when coarse graining with approximate cosets, *i.e.* it is lost. Thus the approximate CRT is still being done, albeit the approximate cosets are implemented differently, which doesn't change the overall algorithm at the macroscopic scale.
> > >
> > > Please see our definition of an approximate coset (3.1) and see Fig. 3. The approximate coset generalizes the traditional coset definition. For a subgroup of \( C_n \), \( \langle m \rangle = \{0, m, 2m, \dots\} \) (where \( m \) divides \( n \)), cosets are residue classes mod \( m \), e.g., \( 1 + \langle m \rangle = \{1, 1+m, 1+2m, \dots\} \), the residue-1 class mod \( m \). The approximate coset generalization defines \( \{c, c+v, c+2v, \dots, c+kv\} \) for \( c, v \in C_n \) and \( 1 \leq k \leq n \), allowing translation of \( c \) by \( v \) up to \( k \) steps.
> > >
> > > As an example, considering elements above a threshold in a simple neuron yields an approximate coset. Normalizing a neuron to frequency 1 reorganizes \( C_n = \{0, 1, 2, \dots, n-1\} \) to align with approximate cosets, where the frequency-1 function activates strongly on them. This remapping is critical, as it highlights this structure regardless of the neuron’s original frequency, enabling qualitative analysis (see Figs. 2, 24). However, in our figures the x-axis could better reflect the "order" after remapping, where adjacent x-values within the maximum step count belong to the same approximate coset.
> > >
> > > For pizza neurons, we observe the same qualitative behavior (see Fig. 19). Despite a secondary frequency evident in the pizza neuron’s DFT, remapping reveals behavior identical to a simple neuron normalized to frequency 1, i.e., strong activation on an approximate coset. From this perspective, the secondary frequency becomes irrelevant; the approximate coset framework effectively abstracts it away, making pizzas and clocks equivalent at an abstract level. Obviously pizzas and clocks are different as shown in Zhong et al. but by this coarse-grained definition, they become the same.
> > >
> > > > "I think the authors should explicitly show that the model they use exhibits pizza algorithm -- or at least point to the exact setup and checkpoint that took from [4]."
> > >
> > > We added to the text the filename (previously we just said "pizza model A in Zhong et al.'s paper").

---

### Author Response · Authors · 2024-11-21
**Global Comment to All Reviewers**

We thank the reviewers for their valuable feedback.

We want to stress that the main purpose of our work was to challenge the notion that the universality hypothesis could be false, by exhibiting that existing "counterexamples" of universality are actually universal at a higher level of abstraction. In particular, the emergent "clock" and "pizza" circuits, which were previously interpreted as distinct circuitry and certificates of the falsehood of universality, can in fact be understood as simple algebraic circuits operating on cosets much like the Chinese Remainder Theorem (CRT). Indeed, our experiments validate that over 400k neural network initializations, all solutions found act by identifying coset membership and superimposing the answers in a CRT-like way. As such, our main claim is that the universality hypothesis has not been conclusively refuted and should still be a topic of great interest.

We provide strong evidence backing our claim through a mathematical model that aligns with all relevant results in the literature, as well as our own quantitative and qualitative experimental findings. Critically, this model allowed us to derive a novel bound that's similar in spirit to a Probably Approximately Correct (PAC) bound, giving scaling of $\mathcal{O}(log(n))$ for the number of frequencies *i.e.* clusters computing cosets) that are learned. Since submission, we trained 600k networks, with 512 neurons, 2048 neurons, and 8096 neurons, totaling over $10^6$ neural networks that we've analyzed. All known experimental results continue to agree with the bound. Finally, please note that the mathematical models in the literature predict $\mathcal{O}(n)$ frequencies are required (not matching experimental results), but ours predicts $\mathcal{O}(log(n))$, tightly matching all experimental results in the literature on this topic. Please see Figures 29, 30 in the appendix of the revised paper for scaling experiments.

Again: all experimental results in the literature, with different numbers of layers, architectures, hyperparameters, and seeds, match our bound. We are not aware of any other work that proposes an interpretation for this problem that can explain so many neural networks, trained with so many different conditions throughout the literature. The interpretation of cosets and approximate cosets, and approximate CRT is the most general interpretation yet.

To make the above goals clearer, we revised the abstract, adjusted the title, and added a first-page figure to illustrate how our work abstracts the network into different computational units. Additionally, we clarified our inclusion of Zhong et al. [1], by adding Appendix B.1 showing that circular and non-circular embeddings are correctly predicted by our paper's modeling: principal components from the same cluster yield circular embeddings, while those from different clusters create non-circular embeddings. We demonstrate this by clearly showing which cluster the principal component came from by adding a DFT of the PCA. In Appendix B.1.1 we show that pizza models use the approximate CRT.

Why is this important? We argue that resolving conflicts in simple settings like modular addition (and finite groups more generally) is a *necessary* step toward understanding the universality of neural networks across tasks. Moreover, uncovering universal mechanisms may be a question of operating at the correct scale, thus these simple tasks offer a fruitful testbed.

[1] Zhong et al., "The Clock and the Pizza: Two Stories in Mechanistic Explanation of Neural Networks" (2023)

---

### Comment · Area_Chair_B9FQ · 2024-11-23
**Discussion period ending soon**

Dear Reviewers,

As we near the end of the discussion period, this is your last chance to engage.

Please review the authors' replies and feedback from other reviewers. If any concerns remain, request clarifications. The authors have one more day to respond before the reviewer-only discussion week begins.

Thank you for your efforts.

Best regards,
Area Chair

---

### Author Response · Authors · 2024-11-29
**A guide to the revision**

We thank the reviewers for their insightful discussions! They helped us clarify the paper. Little information has been added. We focused on improving presentation and addressing reviewer feedback.
- We defined approximate cosets earlier (Section 3) and added a figure visualizing them (Fig. 3)
- We moved tangential content (e.g., representation projections, embedding weight histograms) to the appendix
- We moved the Chinese Remainder Theorem (CRT) from the appendix to the paper (Section 3.1), clarifying its relevance. We added an example of CRT and detailed the CRT algorithm the network learns (Figs 3, 6), emphasizing how the network intersects $\mathcal{O}(\log(n))$ approximate cosets.
- We corrected an error in Fig. 30 [now Fig. 31]: earlier results fixed the network width at 500 neurons, causing under-parameterization when the modulus reached 499, obfuscating the trend. Using Morwani et al.’s ~$7n$ neurons, the updated plot shows clear logarithmic scaling, now emphasized by a log-scaled x-axis cementing our model's prediction.

On Gromov and Morwani et al.: we realized their models and results of $\frac{n-1}{2}$ assume no biases. We trained with biases and observed the emergence of $\mathcal{O}(\log(n))$ frequency clusters, revealing the yet unstated behavior: adding biases causes $\mathcal{O}(\log(n))$ frequencies. This distinction highlights how our work differs from Gromov's and Mowani et al.'s, addressing confusion about novelty compared to these works. We present the first theoretical model that matches experimental results for feature counts in networks with biases. This model and our experiments suggest that more expressive networks, *i.e.* with biases can find more feature efficient solutions -- using $\mathcal{O}(\log(n))$ frequencies instead of $\frac{n-1}{2}$.

We revisit the universality hypothesis and advance interpretability research by showing that low-level circuit differences (clock vs. pizza) do not imply a different algorithm is learned. Using Zhong et al.’s pizza model weights, we qualitatively connect the "clock" (Figs. 6 and 32) and "pizza" (Figs. 18 and 19) interpretations as variations of the same approximate CRT algorithm (Figs. 6, 18, 32 show cluster contributions to logits) and show the preactivations of neurons in clocks (Fig. 2) and pizzas (Figs 18, 19) are effectively equivalent. This surprising result suggests that cosets may underlie finite-group tasks more broadly (Conjecture 2) as the least likely place to expect cosets to appear is on cyclic groups of prime order, as they have no cosets in the strict sense.

---

### Meta-Review · Area_Chair_B9FQ · 2024-12-09

**Metareview:**

The paper considers the problem of understanding the solution identified by a neural network learning modular addition problems. The authors analysed the problem at multiple timescales, from single neuron to the entire network and uncovered non-trivial solution implemented by the classifier. In particular they identified "simple" and "fine-tuning" neurons and showed that the network implements an approximation of the Chinese Remainder Theorem by grouping together neurons.

**Additional Comments On Reviewer Discussion:**

The reviewers agreed on the value of the results and analysis presented in this paper. During the discussion phase, the authors provided several additional results that partially addressed the reviewers' concerns. However, clarity and presentation remain major issues. Several key concepts and arguments were not sufficiently explained, leaving important aspects unclear. Reviewers provided valuable suggestions to improve the clarity of the paper in future iterations.

Another significant concern is that while the authors added extensive new material to the appendix, this content was not well integrated into the main text. As a result, it made the paper harder to follow and, in some cases, introduced more confusion rather than resolving it. The appendix contains valuable content but lacks proper context and integration with the core arguments, making it feel disconnected and underexplained.

I strongly recommend the authors resubmit, ensuring that the new material is properly incorporated into the main body of the paper. Additionally, the appendix should be complemented with more explanations to support the figures and results. The current version feels unpolished, with much of the paper consisting of difficult-to-interpret figures that lack the necessary supporting text.

---

### Decision · Program_Chairs · 2025-01-22

Reject